# Geodesic geometry of 2+1-D Dirac materials subject to artificial, quenched gravitational singularities

**Seth. M. Davis[1*] and Matthew. S. Foster[2]**

**1** Condensed Matter Theory Center and Joint Quantum Institute, Department of Physics, University of Maryland, College Park, MD 20742, USA
**2** Department of Physics and Astronomy and Rice Center for Quantum Materials, Rice University, Houston, Texas 77005, USA

* smdavis1@umd.edu

## Abstract

The spatial modulation of the Fermi velocity for gapless Dirac electrons in quantum materials is mathematically equivalent to the problem of massless fermions on a certain class of curved spacetime manifolds. We study null geodesic lensing through these manifolds, which are dominated by curvature singularities, such as *nematic* singularity walls (where the Dirac cone flattens along one direction). Null geodesics lens across these walls, but do so by perfectly collimating to a local transit angle. Nevertheless, nematic walls can trap null geodesics into stable or metastable orbits characterized by repeated transits. We speculate about the role of induced one-dimensionality for such bound orbits in 2D dirty *d*-wave superconductivity.



## Contents



# 1 Introduction

Many of the most important quantum materials discovered in the past several decades feature electrons, confined to two spatial dimensions, with effective ultrarelativistic band structures. Massless 2D Dirac electrons arise as quasiparticles in the $d$-wave cuprates [1], in monolayer graphene [2], as surface states of bulk topological insulators [3], and in twisted bilayer graphene [4]. Massless Dirac or Majorana quasiparticles are also predicted to form at the surface of topological superfluids and superconductors [5–7].

   Recently, the focus has begun to shift from discovering Dirac materials to precisely manipulating them. In twisted bilayer graphene, for example, the moiré potential flattens the Dirac cones near the magic angle, facilitating Mott insulating and superconducting phases [4]. Since massless Dirac carriers are a fermionic analogue of photons, an interesting question is whether gravitational effects like lensing or trapping behind an event horizon can occur with suitable modifications. *Artificial* quenched gravity (QG) can arise whenever a static source couples to components of the Dirac-electron stress tensor [8]. Coupling to the off-diagonal time-space components breaks time-reversal symmetry ($T$) (as in the Kerr metric [9]) and induces a "tilt" in the Dirac cone [10–13], an effect that is realized in type-II Weyl semimetals [14]. By contrast, we focus here on a static coupling to the spatial-spatial components of the stress tensor that preserves $T$, but modulates the components of the Dirac velocity (the effective "velocity of light"), see Fig. 1.

   While it is possible to deform the velocity in normal Dirac materials like graphene using strain [15,16], it is particularly natural in Dirac superconductors (SCs). For example, a charged impurity placed on the surface of class DIII topological SC is predicted to isotropically steepen or flatten the Dirac cone of the surface Majorana fluid [17]. In 2D $d$-wave SCs such as the cuprates, a modulation of the pairing amplitude translates into a *nematic* deformation of the Dirac cone (along the Fermi surface). Nematicity and emergent one-dimensionality have been argued to play a key role in the physics of high-temperature superconductivity [18–21].

   *Random* quenched gravity arises when the Dirac velocity modulation occurs due to disorder ("quenched gravitational disorder," QGD). Nanometer-scale inhomogeneity observed by tunneling into BSCCO [22] could imply that QGD plays a role in high-temperature super-

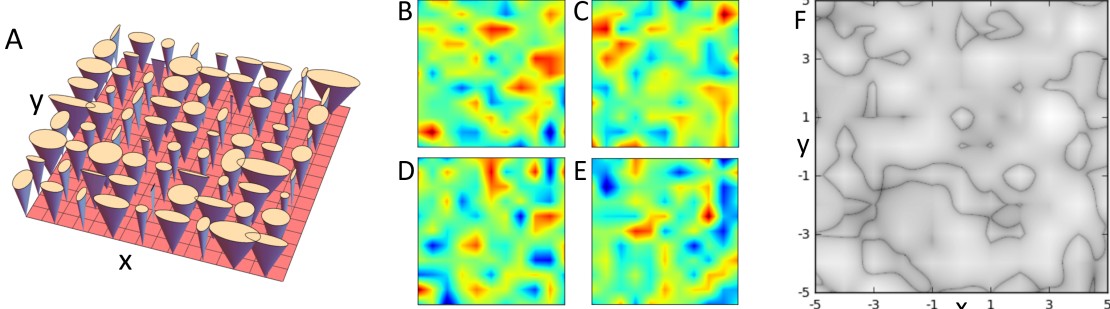

Figure 1: Introduction to the gravitational view of Dirac-carrier velocity modulation. A: Cartoon depicting a random spatial modulation of a 2D Dirac cone (QGD—see text). B-E: Heat maps on a spatial $[-5, 5] \times [-5, 5]$ grid, depicting four randomly-generated (Gaussian) disorder potentials, corresponding to the velocity components $v_a^b(\mathbf{x})$; these couple to the spatial components of the Dirac-electron stress tensor in Eq. (1). F: Heat map depicting the gravitational time-dilation factor, relative to the flat case [a proxy for curvature, see Eq. (25) and surrounding discussion], for the manifold corresponding to the disorder potentials in B-E. Note the visible domain walls corresponding to 1D nematic curvature singularities.

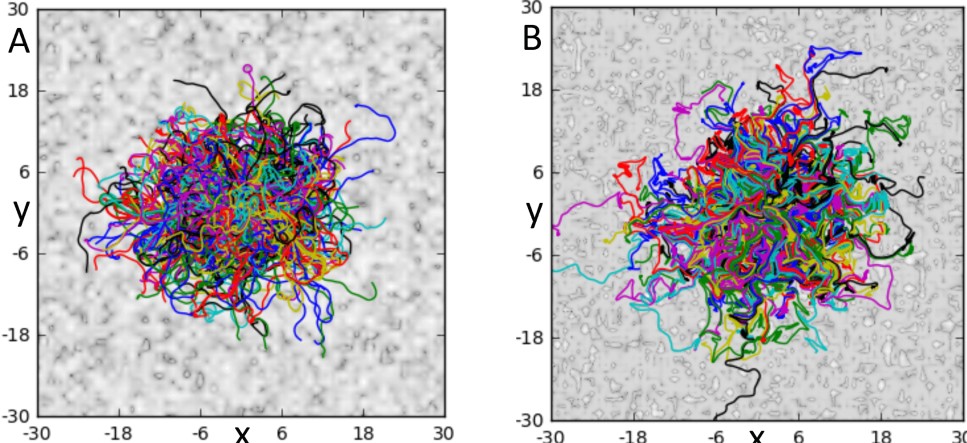

Figure 2: Generic null geodesic trajectories in the spatial plane for purely isotropic or purely nematic QGD. As discussed in the text, we consider quenched random space-times with a special *temporal flatness* condition, which means that 2+1-D Dirac carriers are modulated by perturbations that artificially mimic gravity, but in samples defined in physically flat spacetime. It means that physical distances in a solid-state realization correspond to the Euclidean measure in the plane, rather than to the geodesic one. A: Null geodesics for a purely isotropic QGD realization. Trajectories with different initial conditions (initial position and launch angle) appear with different colors. Note the qualitative resemblance to uncorrelated 2D random walks (diffusion). B: Null geodesics for a purely nematic QGD realization. Nearby geodesics are highly correlated in their direction, and tend to exhibit near-retracing orbits that bounce back and forth along nematic curvature singularity contours. Note that only in case B do singularities arise along 1D curves (domain walls).

conductivity; it has recently been demonstrated that increasing disorder can raise the critical temperature in these materials [23]. QGD might also arise due to twist disorder in bilayer graphene [24, 25]. The physics of QGD has only recently been investigated theoretically. Exact diagonalization studies of a 2D Dirac cone subject to different varieties of QGD revealed a surprisingly robust incarnation of quantum criticality. In Ref. [17], nematic QGD was shown to produce an entire spectrum of quantum-critical single-particle wave functions, with universal critical spatial fluctuations analogous to those found at an Anderson metal-insulator transition. Spectrum-wide quantum criticality has also been observed in non-gravitational models of topological superconductor surface states, where it was linked to quantum Hall plateau transitions [26–28].

Motivated by the prospects for inducing gravitational effects in 2D Dirac materials, and by the numerical observation of robust quantum criticality induced by QGD [17], this paper investigates the geometry of 2+1-D Lorentzian manifolds with quenched gravitational singularities, corresponding to gravitationally modulated Dirac materials. We study the semiclassical limit of massless Dirac carriers lensing through static gravitational landscapes by computing null geodesic trajectories. In classical general relativity, null geodesics give the trajectories followed by massless particles. More generally, null geodesics play a crucial role in informing the solution to the wave equation on curved manifolds [29]. Geodesics can form the basis for a sensible semiclassical expansion of the Dirac equation, which governs a consistent single-particle relativistic quantum mechanics (unlike the Klein-Gordon equation). These points are formalized by the fact that null geodesics are exactly the bicharacteristics for the Dirac equation on a Lorentzian manifold. Bicharacteristics determine the propagation of discontinuities of partial differential equations and very generally correspond to a "geometric optics" view-

point of a generic field equation [30]. We emphasize that geodesic-based approaches have been employed sucessfully in condensed matter contexts – in particular, we note that conductivity corrections due to notrivial "geodesic scattering" have been studied previously in the context of 3D TI surface states [31].

To be precise, we consider only *artificial* gravitational potentials, i.e. perturbations that mimic gravity by coupling to the stress tensor, but for Dirac electrons propagating through physically flat spacetime. Technically, this translates into a special *temporal flatness* condition, which simplifies the metric and the analysis of null geodesics. Temporal flatness provides a restriction on the types of spacetime geometries that can be realized with emergent Dirac-type Hamiltonians existing in a physically flat spacetime. We emphasize that many of the famous spacetime metrics of general relativity (e.g. the Schwarzschild metric) do *not* satisfy temporal flatness. This also means that there is a preferred coordinate system that measures physical Euclidean distances in the plane; geodesic distances are "experienced" by the Dirac carriers, but would not be easily extracted from an experiment. Examples of the static gravity studied here include nematic deformations of the Dirac cone, as can arise from spatial inhomogeneity in the pairing potential of a $d$-wave SC, or isotropic flattening of the Majorana cone due to impurities at the surface of a topological SC [17]. By contrast, our results are not applicable to a macroscopically curved sheet of graphene.

In the gravitational language, curvature singularities arise whenever one or both components of the Dirac carrier velocity vanish. We study two different types of singular loci. Nematic singularities occur when only one component of the Dirac velocity passes through zero, and arise along 1D curves in a $d$-wave superconductor whenever the pairing amplitude changes sign (i.e., separates pairing domains with a $\pi$ phase shift). We also study isotropic curvature singularities, where the entire Dirac cone is flattened at a point.

We find that there are strong qualitative differences between the geodesics corresponding to nematic and isotropic QGD, as shown in Fig. 2. Moreover, we show that null geodesics are profoundly influenced by isotropic and nematic curvature singularities, though these give rise to different effects. Isotropic singularities are strongly attractive and asymptotically capture geodesics that pass sufficiently close. Conversely, nematic singularities do not capture geodesics; instead, they drive all impinging geodesics to a unique velocity, dependent only on the disorder potentials at the singularity, at which the geodesics are allowed to pass through

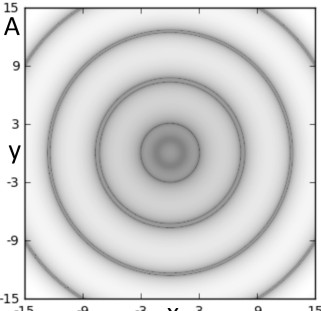 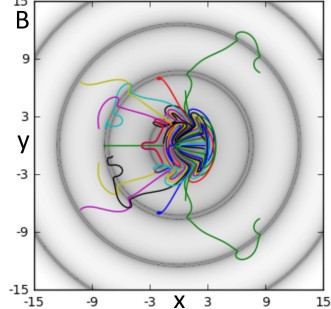 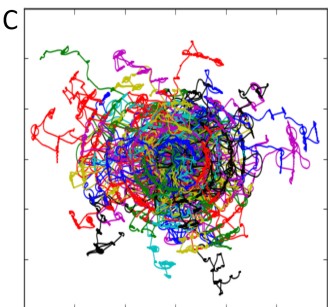

Figure 3: Example depicting metastable null geodesic bound states along nematic singularities. The spacetime manifold is defined by $v_D = \cos[0.2\pi r]x$, $v_N = \cos[0.2\pi r]y$ in the purely nematic model of Eqs. (51a)–(51b). A: Time dilation heat map of the manifold [Eq. (25)], depicting concentric circular contours of curvature singularity. B: Null geodesics shortly after release from the origin. We see that the geodesics tend to travel along the singular contours. C: Null geodesics in the long-time limit, showing that the geodesic dynamics are dominated by metastable orbits along these singularities.

the singularity. We call this effect *geodesic collimation*. The collimation angle is simply determined by the local dispersive direction of the Dirac cone at the singularity wall.

Although the null geodesics do not "stick" to singular nematic contours, there is a horizon effect due to the latter. In particular, nematic singularity walls can produce stable and metastable gravitationally bound orbits, wherein null geodesics repeatedly lens back and forth across the wall (at the local geodesic collimation angle). This is illustrated for a "nematic circular billiard" spacetime shown in Fig. 3.

## 1.1 Outline

Our manuscript is organized as follows. Sec. 2 introduces the 2D Dirac Hamiltonian for artificial quenched gravity. Mapping this to the covariant formulation of massless fermions on curved spacetime, we extract an associated metric. We discuss symmetries and describe the nature of curvature singularities that can arise in this spacetime. In Sec. 3, we develop the geodesic equation and reformulate it several times to gain insight into the geometric properties of geodesics, especially in the vicinity of curvature singularities. The most useful formulation employs the projection of the tangent vector onto the local dreibein. Sec. 4 introduces the purely isotropic and purely nematic submodels, which allow separate study of nematic and isotropic Dirac cone modulation. In Sec. 5 we present an array of simple, highly-symmetric but singular spacetime geometries that admit analytical solutions and give a window into properties of the null geodesic interactions with singularities. Finally, Sec. 6 revisits the spacetime geometries of Sec. 5 from a fully quantum perspective to provide some intuition for the ways in which geodesics can approximate the true quantum picture. Sec. 7 summarizes our results and discusses future directions.

Many supporting details and calculations are relegated to appendices. Appendix A summarizes various useful reformulations of the geodesic equation. Appendix B explains that the geodesic dynamics do not depend qualitatively on the length scale of a QGD potential, and Appendix C introduces two other submodels that aid connection with previous work [17]. In addition to the analytical calculations described in Sec. 5, we compute numerical results for regular and random quenched gravitational potentials, exhibited in various figures displayed throughout the paper.

## 2 Model and analogy to gravitation

### 2.1 Hamiltonian

Our focus is a 2+1-dimensional massless Dirac model in the presence of *artificial* quenched gravity: perturbations that mimic gravity by coupling to the spatial-spatial components of the stress tensor, and which preserve time-reversal symmetry. These flatten, steepen, or rotate the components of the Dirac velocity. Despite the spatial modulation of the Dirac cone, we assume that the Dirac electrons propagate through physically flat spacetime; in practice, this means that we exclude transport across a curved 2D sample (such as a corrugated graphene sheet). As explained in the Introduction 1, this is hardly a restriction in the context of 2D Dirac materials. As an example, for Dirac or Majorana quasiparticles that arise at the boundary of 3D topological or in 2D $d$-wave superconductors, inhomogeneity due to charged impurities or modulation of the pairing gap can both manifest as quenched gravitational coupling to the stress tensor [17].

Our model is defined by the Hamiltonian

$$\mathcal{H} = -\frac{i}{2} \sum_{a,b=1,2} \int d^2\mathbf{x}\, v_a^b(\mathbf{x}) \left[ \bar{\psi}(\mathbf{x})\, \hat{\sigma}^a \overleftrightarrow{\partial_b}\, \psi(\mathbf{x}) \right], \tag{1}$$

where $\mathbf{x} = \{x, y\}$ are Cartesian coordinates that measure physical Euclidean distances in the sample plane, $\psi = \psi_\sigma$ is a two-component spinor, $\hat{\sigma}^{1,2}$ are Pauli matrices, and the double-directed derivative is defined by $(f \overleftrightarrow{\partial} g) \equiv f(\partial g) - (\partial f)g$. The functions $\{v_a^b(\mathbf{x})\}$ couple to the spatial components of the Dirac-electron stress tensor $T^a{}_b$. In the unperturbed limit, we can take $\{v_1^1 = v_2^2 = 1,\ v_1^2 = v_2^1 = 0\}$.

We will find it useful to define the "disorder vectors":

$$\mathbf{v_j}(\mathbf{x}) = \begin{bmatrix} v_1^j(\mathbf{x}) \\ v_2^j(\mathbf{x}) \end{bmatrix}, \tag{2}$$

which will let us write the action of our fermion theory as

$$\mathcal{S} = i \int dt \int d^2\mathbf{x} \left\{ \bar{\psi}(t,\mathbf{x})\partial_t \psi(t,\mathbf{x}) + \frac{1}{2} \sum_{a,b=1,2} v_a^b(\mathbf{x}) \left[ \bar{\psi}(t,\mathbf{x})\, \hat{\sigma}^a \overleftrightarrow{\partial_b}\, \psi(t,\mathbf{x}) \right] \right\}, \tag{3a}$$

$$= i \int dt \int d^2\mathbf{x}\, \bar{\psi}(t,\mathbf{x}) \begin{pmatrix} \partial_t + [\mathbf{v_1}(\mathbf{x}) \cdot \hat{\boldsymbol{\sigma}}]\partial_1 + \frac{1}{2}[\partial_1 \mathbf{v_1}(\mathbf{x}) \cdot \hat{\boldsymbol{\sigma}}] \\ + [\mathbf{v_2}(\mathbf{x}) \cdot \hat{\boldsymbol{\sigma}}]\partial_2 + \frac{1}{2}[\partial_2 \mathbf{v_2}(\mathbf{x}) \cdot \hat{\boldsymbol{\sigma}}] \end{pmatrix} \psi(t,\mathbf{x}), \tag{3b}$$

where $\hat{\boldsymbol{\sigma}} = [\hat{\sigma}^1, \hat{\sigma}^2]^T$. In going from Eq. (3a) to Eq. (3b), we integrate by parts to remove the double-directed derivative in Eq. (3a), which we see gives rise to a spatially-dependent imaginary vector potential in the Lagrangian. Hermiticity requires that such terms exist to balance the spatially modulated Dirac velocity; preserving Hermiticity is crucial, since non-Hermitian versions of the spatial stress tensor arise instead for reparameterization ghosts in 2+0-D [32]. The Hermitian counterterms in Eq. (3b) form the *spin connection* of the covariant Lagrangian for a theory of massless fermions on a curved spacetime manifold.

It will also be helpful to define $[\mathbf{u_1}, \mathbf{u_2}] = [\mathbf{v_1}, \mathbf{v_2}]^T$, which allows us write the Hamiltonian as

$$\mathcal{H} = \int d^2\mathbf{x}\, \bar{\psi} \left( -i \sum_j \hat{\sigma}^j \left[ (\mathbf{u_j} \cdot \boldsymbol{\nabla}) + \frac{1}{2}(\boldsymbol{\nabla} \cdot \mathbf{u_j}) \right] \right) \psi. \tag{4}$$

In momentum space, the Dirac cone is spanned by the vectors $\{\mathbf{u_j}\}$.

## 2.2 Mapping to gravity

We pursue a gravitational analogy to shed light on the Dirac Hamiltonian in Eq. (1). Using the vielbein formalism [9], the action for a system of massless Dirac fermions on a 2+1-dimensional spacetime is given by [33]

$$\mathcal{S} = \int \sqrt{|g|}\, d^3x\, \bar{\psi}(x) E_A^\mu(x)\hat{\gamma}^A \left[ i\partial_\mu - \frac{1}{2}\omega_\mu^{BC}(x)\hat{S}_{BC} \right] \psi(x), \tag{5}$$

where $x = \{t, x, y\}$, $\mu \in \{0, 1, 2\}$ is a coordinate index, $A, B, C \in \{0, 1, 2\}$ are local Lorentz indices, the 2×2 Dirac matrices satisfy the Clifford algebra

$$\hat{\gamma}^A \hat{\gamma}^B + \hat{\gamma}^B \hat{\gamma}^A = -2\eta^{AB}\,\hat{1},$$

$\eta^{AB} \to \text{diag}(-1, 1, 1)$ is the flat Minkowski metric, $g_{\mu\nu}$ is the spacetime metric and $g$ is its determinant. Further, $E_A^\mu$ is the *dreibein*, central to the tetrad (here: "triad") formalism, defined by the relation

$$\eta^{AB} E_A^\mu(x) E_B^\nu(x) = g^{\mu\nu}(x). \tag{6}$$

Finally, $\hat{S}_{BC}$ generates local Lorentz transformations on the spinor index of $\psi$, and $\omega_\mu^{BC}$ is the *spin connection*, defined by

$$\omega_{\mu B}^A = E_\nu^A \Gamma_{\mu\lambda}^\nu E_B^\lambda - (\partial_\mu E_\lambda^A) E_B^\lambda. \tag{7}$$

Our goal is to identify a spacetime metric $[g^{\mu\nu}(\mathbf{x})]$ such that Eqs. (3b) and (5) are identical. After shifting $\bar{\psi} \to \bar{\psi}\hat{\gamma}^0$ in Eq. (5) [such that $\bar{\psi} \leftrightarrow \psi^\dagger$, implicitly assumed in the "non-relativistic" notation of Eq. (3b)], we can identify $\hat{\gamma}^0\hat{\gamma}^a = \hat{\sigma}^a$. Consistency between Eqs. (3b) and (5) requires setting $E_{A\neq0}^0 = 0$, due to the absence of time-space mixing terms. For static $\{\mathbf{v}_j(\mathbf{x})\}$, this is equivalent to enforcing time-reversal symmetry. Further, as explained above and in the Introduction, the Dirac electrons are assumed to propagate through physically flat 2+1-D spacetime, with effective gravitation arising solely due to the spatial variation in $v_a^b(\mathbf{x})$. Then, the coefficient of the time-derivative term in Eq. (5) can be chosen equal to one, a condition that we call *temporal flatness*,

$$\sqrt{|g|} E_0^0 \equiv 1. \tag{8}$$

With this choice, the Cartesian coordinates $\mathbf{x}$ in Eqs. (3b) and (5) measure physical Euclidean distances in the plane. Temporal flatness allows the identification of the disorder potentials directly in terms of the dreibein,

$$v_a^b = \frac{E_a^b}{E_0^0}. \tag{9}$$

We may then construct the metric in terms of $v_a^b$ and $E_0^0$ via Eq. (6). If we bring the dreibein in line with the potentials in Eq. (9), then the spin connection will match the imaginary vector potential terms in Eq. (3b).

To determine $E_0^0$, we take the determinant of the metric and again invoke temporal flatness to compute

$$1 = \frac{1}{(E_0^0)^2|g|} = [E_2^1 E_1^2 - E_1^1 E_2^2]^2 = (E_0^0)^4 (\mathbf{v_1} \times \mathbf{v_2})^2. \tag{10}$$

We thus find that Eq. (5) is equivalent to the Hamiltonian system in Eq. (1) if the dreibein and spacetime metric are given by the mapping dictionary

$$E_0^0 = \frac{1}{\sqrt{|\mathbf{v_1} \times \mathbf{v_2}|}}, \tag{11a}$$

$$E_A^\mu \to \frac{1}{\sqrt{|\mathbf{v_1} \times \mathbf{v_2}|}} \begin{bmatrix} 1 & 0 & 0 \\ 0 & v_1^1 & v_2^1 \\ 0 & v_1^2 & v_2^2 \end{bmatrix}, \tag{11b}$$

$$g_{\mu\nu} \to \frac{1}{|\mathbf{v_1} \times \mathbf{v_2}|} \begin{bmatrix} -(\mathbf{v_1} \times \mathbf{v_2})^2 & 0 & 0 \\ 0 & |\mathbf{v_2}|^2 & -\mathbf{v_1} \cdot \mathbf{v_2} \\ 0 & -\mathbf{v_1} \cdot \mathbf{v_2} & |\mathbf{v_1}|^2 \end{bmatrix}. \tag{11c}$$

The result in Eq. (11c) defines the quenched gravitational metric. This metric is quite general, although it has three important structural properties that constrain the geometry: (1)

It is everywhere block-diagonal in time and space [a consequence of time-reversal symmetry for static potentials $\{\mathbf{v}_j(\mathbf{x})\}$]. (2) The temporal flatness condition [Eq. (8)], equivalent here to the condition $g_{00} = \det(g_{\mu\nu})$, implies an everywhere flat spacelike area measure $1 \times dx\, dy$. This is different from (e.g.) the Schwarzschild metric. (3) The metric is time-independent [$\partial_t g_{\mu\nu} = 0$]. We note that if one wanted to consider time-dependent gravitational disorder by allowing for explicit time-dependence of the disorder vectors $\{\mathbf{v}_j\}$, only the last of these conditions is removed [allowing for the generalization presented in Eq. (36)]. We also note that metric is expressed entirely in terms of the relative geometry of the disorder vectors, a fact that will be important for establishing the invariance of the geodesic dynamics to pseudospin rotations in Sec. 2.4.

Our theory can thus be studied in two different settings. On one hand, it is an effective low-energy Dirac theory due to perturbations that couple to spatial-spatial stress tensor components in a condensed matter system. On the other hand, we can study it as a theory of free massless fermions on a corresponding curved spacetime.

## 2.3 Curvature and singularities

The metric in Eq. (11c) becomes ill-defined at points where the cross-product $\mathbf{v}_1 \times \mathbf{v}_2$ vanishes. This condition corresponds to a failure of the temporal flatness condition [Eq. (10)], divergence of the dreibein [Eq. (11b)], and the non-invertibility of the inverse metric. The Ricci scalar curvature takes the form

$$R = \frac{N\left(v_a^b, \partial v_c^d\right)}{|\mathbf{v}_1 \times \mathbf{v}_2|^3}, \tag{12}$$

where $N\left(v_a^b, \partial v_c^d\right)$ is a (complicated) homogeneous quadratic polynomial in spatial derivatives of the disorder-vector components. While it is possible for the numerator to vanish so as to give finite curvature at a point where $\mathbf{v}_1 \times \mathbf{v}_2 = 0$, we will generically find curvature singularities along the sets defined by this condition.

We can characterize singularities in terms of Dirac cone geometry: at a point where $\mathbf{v}_1 \times \mathbf{v}_2 = 0$, we have

$$\begin{aligned}
\mathbf{v}_1 &= \cos\theta^* \, \mathbf{v}, \\
\mathbf{v}_2 &= \sin\theta^* \, \mathbf{v},
\end{aligned} \tag{13}$$

for some $\mathbf{v} \equiv [v_1, v_2]^T$ and an angle $\theta^* = \arctan(|\mathbf{v}_2|/|\mathbf{v}_1|)$. In the notation of Eq. (4), $\mathbf{u}_1 = v_1 \hat{\boldsymbol{\theta}}^*$ and $\mathbf{u}_2 = v_2 \hat{\boldsymbol{\theta}}^*$, where $\hat{\boldsymbol{\theta}}^* = [\cos\theta^*, \sin\theta^*]^T$. It follows that

$$\mathcal{H} = \int d^2\mathbf{x}\, \bar{\psi}\left[-i(\mathbf{v} \cdot \hat{\boldsymbol{\sigma}})\partial_{\theta*} + \text{S.C.}\right]\psi, \tag{14}$$

where S.C. is the spin connection term. At the singularity, the energy only depends on the derivative of the field in the direction of $\hat{\boldsymbol{\theta}}^*$: there is a flat band in the perpendicular direction, forming a "Dirac canyon."

A singularity can thus be partially characterized in terms of the angle $\theta^*$ and the vector $\mathbf{v}$, as defined above. At a singularity point, we have a flattening of *at least one* axis of the Dirac cone, in the direction perpendicular to $\hat{\boldsymbol{\theta}}^*$. We see that there are two types of possible curvature singularities: *nematic* singularities correspond to nonzero $\mathbf{v}$ [Eq. (13)] and give rise to a local Dirac canyon, while *isotropic* singularities correspond to $\mathbf{v} = 0$, and locally flatten the entire Dirac cone. We note that isotropic deformations of the Dirac cone can only produce isotropic singularities. On the other hand, nematic singularities can only be formed by the breaking of rotational symmetry of the electron band structure.

We can gain more insight with some topological reasoning. The quantity $\mathbf{v_1} \times \mathbf{v_2}$ can vary continuously with $\mathbf{x}$, taking on both negative and positive values. Thus, the regular singularities will generally form 1-manifolds that act as domain walls, partitioning the plane into regions of $\mathbf{v_1} \times \mathbf{v_2} > 0$ and $\mathbf{v_1} \times \mathbf{v_2} < 0$. Even at a singular point $|\mathbf{v}| \geq 0$, and so isotropic singularities will generally arise only at isolated points.

We will see in later sections that both flavors of singularity strongly impact geodesic behavior. Geodesics that collide with an isotropic singularity are arrested and remain captured for the rest of time. These isotropic singularities also turn out to exert a strong pull on nearby geodesics. Conversely, geodesics that collide with a nematic singularity pass through in finite time; they are all driven to pass through the singularity in the direction $\hat{\boldsymbol{\theta}}_*$ and at the speed $|\mathbf{v}|$, a singularity-induced *geodesic collimation* effect. We stress that, unintuitively, this fixing of both the geodesics' direction and speed does *not* uniquely define the geodesic.

## 2.4   Pseudospin rotations

Before moving on to a study of the spacetime manifolds defined by the metric in Eq. (11c), we pause to consider the properties of the quantum theory [Eq. (3b)] under a local pseudospin rotation. We claim that the *dynamics* of the theory are invariant under a local $U(1)$ pseudospin symmetry.

Specifically, let the unitary transformation

$$\mathcal{U}(\mathbf{x}) = \exp\left[\frac{i}{2}\theta(\mathbf{x})\hat{\sigma}^3\right] \tag{15}$$

encode the in plane rotation $\mathbf{v} \to \hat{R}(\mathbf{v})$ via the canonical SU(2) $\to$ SO(3) double cover. That is,

$$\mathcal{U}^\dagger[\mathbf{v} \cdot \hat{\boldsymbol{\sigma}}]\mathcal{U} = \hat{R}(\mathbf{v}) \cdot \hat{\boldsymbol{\sigma}}, \tag{16}$$

where the rotation operator $\hat{R}$ is given by

$$\hat{R}(\mathbf{v}) = \cos\theta\mathbf{v} - \sin\theta\mathbf{v}^\perp. \tag{17}$$

(Our convention is that $\mathbf{v}^\perp = [v_2, -v_1]^T$.)

The unitary fermion field transformation $\psi \to \mathcal{U}\psi$ sends the action [Eq. (3b)] to

$$\mathcal{S} = \int dt \int d\mathbf{x}\, \bar{\psi}(t,\mathbf{x}) \left(\begin{array}{l} i\partial_t + [\hat{R}(\mathbf{v_1}) \cdot \hat{\boldsymbol{\sigma}}]i\partial_x + \dfrac{i}{2}[\partial_x \hat{R}(\mathbf{v_1}) \cdot \hat{\boldsymbol{\sigma}}] \\[2mm] \quad + [\hat{R}(\mathbf{v_2}) \cdot \hat{\boldsymbol{\sigma}}]i\partial_y + \dfrac{i}{2}[\partial_y \hat{R}(\mathbf{v_2}) \cdot \hat{\boldsymbol{\sigma}}] \end{array}\right) \psi(t,\mathbf{x}). \tag{18}$$

While the action is not invariant, the new theory is not qualitatively different from the old. The disorder vectors have been rotated through the same angle and their relative geometry is preserved. Since the metric for the corresponding spacetime manifold [Eq. (11c)] depends only on the lengths and relative angles of the disorder vectors, it is explicitly invariant under the transformation. It follows that the geodesic dynamics are invariant as well.

The quantum dynamics are also invariant under the transformation. To see this, note that the quantum states of the original theory can be recovered from knowledge of the quantum states of the pseudospin-rotated theory by enacting the inverse pseudospin rotation [$\mathcal{U}^\dagger(\mathbf{x})$] on the eigenstates. The same can be said of the time-dependent wave function. Since the time-dependent wave functions are related by a unitary pseudospin rotation, the corresponding time-dependent density functions are identical.

## 3 Geodesics

The main focus of this paper is the study of the geodesics on the manifolds defined by the quenched gravitational metric in Eq. (11c). In this section, we introduce the geodesic equation and reformulate it into a more manageable form that allows an analytical understanding of the effects of curvature singularities, and also facilitates efficient numerical evaluation.

### 3.1 Geodesic equation

The geodesic equation is given by the second order ODE [9]

$$\frac{d^2 x^\mu}{ds^2} = -\Gamma^\mu_{\alpha\beta}[\mathbf{x}(s)] \frac{dx^\alpha}{ds} \frac{dx^\beta}{ds}, \tag{19}$$

where $s$ is an affine parameter for the curve and $\{\Gamma^\mu_{\alpha\beta}\}$ are the Christoffel symbols derived from the metric [Eq. (11c)]. In our case, these take the form

$$\Gamma^0_{\mu\nu}(\mathbf{x}) = \begin{bmatrix} 0 & \partial_1 & \partial_2 \\ \partial_1 & 0 & 0 \\ \partial_2 & 0 & 0 \end{bmatrix} \frac{1}{2} \log |\mathbf{v_1} \times \mathbf{v_2}|, \tag{20a}$$

$$\Gamma^1_{\mu\nu}(\mathbf{x}) = \begin{bmatrix} \Gamma^1_{00}(\mathbf{x}) & 0 & 0 \\ 0 & \Gamma^1_{11}(\mathbf{x}) & \Gamma^1_{12}(\mathbf{x}) \\ 0 & \Gamma^1_{12}(\mathbf{x}) & \Gamma^1_{22}(\mathbf{x}) \end{bmatrix}, \tag{20b}$$

$$\Gamma^2_{\mu\nu}(\mathbf{x}) = \begin{bmatrix} \Gamma^2_{00}(x) & 0 & 0 \\ 0 & \Gamma^2_{11}(\mathbf{x}) & \Gamma^2_{12}(\mathbf{x}) \\ 0 & \Gamma^2_{12}(\mathbf{x}) & \Gamma^2_{22}(\mathbf{x}) \end{bmatrix}. \tag{20c}$$

The Christoffel symbols $\{\Gamma^\rho_{\mu\nu}\}$ left undefined above are complicated functions of $\mathbf{v}_{1,2}$ and derivatives thereof.

When parameterized by proper time (or an affine parameter), solutions of the geodesic equation [Eq. (19)] have a conserved *spacetime interval*:

$$g_{\mu\nu}[x(s)] \frac{dx^\mu}{ds} \frac{dx^\nu}{ds} \equiv \Delta, \tag{21}$$

where

$$\Delta = \begin{cases} -1 & \text{(``timelike geodesics'')}, \\ 0 & \text{(``null geodesics'')}, \\ +1 & \text{(``spacelike geodesics'')}. \end{cases} \tag{22}$$

Null geodesics correspond to the trajectories followed by massless particles (e.g., light in general relativity). Since we are interested in massless Dirac particles, we will focus on this case.

### 3.2 Temporal first integral

The structure of the quenched spacetime in Eq. (11c) yields a general first integral for the time coordinate along a geodesic. Inserting Eq. (20a) into the geodesic Eq. (19) gives

$$\frac{d^2 t}{ds^2} = -(\partial_1 \log |\mathbf{v_1} \times \mathbf{v_2}|) \frac{dx}{ds} \frac{dt}{ds} - (\partial_2 \log |\mathbf{v_1} \times \mathbf{v_2}|) \frac{dy}{ds} \frac{dt}{ds}. \tag{23}$$

This is integrable and the first integral for time follows:

$$\frac{dt}{ds} = \frac{(E/m)}{\left|\mathbf{v_1}[\mathbf{x}(s)] \times \mathbf{v_2}[\mathbf{x}(s)]\right|}.$$ (24)

Above, $E/m$ is the constant of integration, which we identify as the energy of the geodesic according to an observer at rest at the same location. To see this, note that the standard expression for this is $E/m = -g_{00}[\mathbf{x}(s)](dt/ds)$. The fact that energy is conserved along a generic geodesic is due to the fact that the "at rest" three-vector $\hat{t} \equiv [1,0,0]^T$ gives a global timelike Killing field for the quenched gravitational manifold. For null geodesics, the constant $E/m$ formally diverges, but it can be scaled arbitrarily without affecting the geodesic.

### 3.3 Reparametrization by global time

In light of Eq. (24), it will be useful to define

$$\gamma(\mathbf{x}) \equiv \frac{1}{\left|\mathbf{v_1}(\mathbf{x}) \times \mathbf{v_2}(\mathbf{x})\right|},$$ (25)

which is the *gravitational time-dilation factor*.

Since $dt/ds = \gamma[\mathbf{x}(s)] > 0$ [setting $E/m = 1$ in Eq. (24)], the mapping between the affine parameter $s$ and global time $t$ along a geodesic is invertible. There is thus a well-defined reparametrization of the geodesic in terms of $t$ [$\equiv \mathbf{x}(t)$]. Using Eq. (24), we have ($j \geq 1$)

$$\frac{dx^j}{dt} = \frac{dx^j}{ds}\frac{ds}{dt} = \frac{1}{\gamma[\mathbf{x}(s)]}\frac{dx^j}{ds},$$ (26)

so that tangent vectors of geodesics with respect to the global time coordinate are just spatially-dependent dilations of the original tangent vectors with respect to the affine parameter. The geodesic equation in terms of the global time coordinate is [with $\dot{x}_j \equiv dx_j/dt$]

$$\ddot{x}(t) = -\left[\Gamma^1_{00} + (\Gamma^1_{11} + \partial_1\log[\gamma])\dot{x}^2 + (2\Gamma^1_{12} + \partial_2\log[\gamma])\dot{x}\dot{y} + \Gamma^1_{22}\dot{y}^2\right],$$ (27a)

$$\ddot{y}(t) = -\left[\Gamma^2_{00} + \Gamma^2_{11}\dot{x}^2 + (2\Gamma^2_{12} + \partial_1\log[\gamma])\dot{x}\dot{y} + (\Gamma^2_{22} + \partial_2\log[\gamma])\dot{y}^2\right].$$ (27b)

These equations offer some interesting interpretations. Firstly, the $\Gamma_{00}$ terms appear as potentials in what is effectively a Hamiltonian dynamics problem with many friction-like dissipative terms. We note that the global time reparametrization introduces several new terms combining with the Christoffel symbols, adding new friction-like terms to the geodesic equation, corresponding to the drag-like effects of time-dilation. These dissipative terms play a key role in the geodesic capture by isotropic curvature singularities, which would otherwise be forbidden by conservation of energy. We discuss this again in Sec. 4.1.

### 3.4 Reformulation of the geodesic equation

Introducing notation for the speed [$\sigma \equiv |\dot{\mathbf{x}}|$] and velocity angle [$\theta(t) \equiv \arctan[\dot{y}(t)/\dot{x}(t)]$], we can use Eq. (21) to relate a geodesic's speed, position, energy, and mass. Eqs. (24) and (25) imply that

$$\left(\frac{E}{m}\right)^2\left\{1 - \gamma^2(\mathbf{x})\sigma^2(\mathbf{x})\left[(\mathbf{u_1} \times \hat{\boldsymbol{\theta}})^2 + (\mathbf{u_2} \times \hat{\boldsymbol{\theta}})^2\right]\right\} = -\frac{\Delta}{\gamma(\mathbf{x})},$$ (28)

where again, $\hat{\boldsymbol{\theta}} \equiv [\cos\theta, \sin\theta]^T$ and $[\mathbf{u_1}, \mathbf{u_2}] = [\mathbf{v_1}, \mathbf{v_2}]^T$. Solving instead for the squared-speed of the geodesic, we find

$$\sigma^2_\Delta[\theta, \mathbf{x}] = \frac{1}{\gamma(\mathbf{x})^2}\frac{1}{(\mathbf{u_1} \times \hat{\boldsymbol{\theta}})^2 + (\mathbf{u_2} \times \hat{\boldsymbol{\theta}})^2}\left[1 + \frac{\Delta}{\gamma(\mathbf{x})}\left(\frac{m}{E}\right)^2\right].$$ (29)

In the flat-space limit, these reduce to the familiar equations of special relativity: $E^2(1-\sigma^2) = m^2$ for a timelike geodesic. Note also that if $m = 0$, then $E$ plays no role, reflecting that fact that null geodesics are unaffected by a scaling of the affine parameter. Eq. (29) can be used to rewrite the geodesic equation in an angular formulation; this is presented in Appendix A.

While our focus is on null geodesics, we see that timelike and spacelike geodesics have speed-position relations derived from those of null geodesics by a simple multiplicative factor. From Eq. (29), we can see that while null $[\Delta = 0]$ and tachyonic $[\Delta > 0]$ geodesics have a well-defined speed at every point of the manifold, massive geodesics $[\Delta < 0]$ are restricted from the regions of the manifold with $-\Delta < (E/m)^2 \gamma(\mathbf{x})$.

Equation (29) with $\Delta = 0$ also offers an insight into the null geodesic collimation effect alluded to in Sec. 2.3. At a singularity, the factor $\gamma^{-2}$ necessarily vanishes, pushing the speed of the geodesic towards zero. The geodesic may only pass through the singularity if the denominator in Eq. (29) diverges simultaneously. This can happen only if $\mathbf{u_1}$ and $\mathbf{u_2}$ are parallel (automatic for the singularity), *and* if the velocity vector of the geodesic is driven to point in their common direction, $\hat{\boldsymbol{\theta}} \to \hat{\boldsymbol{\theta}}^*$ [see the discussion around Eq. (14)]. Though it is not obvious from Eq. (29), we will see that all geodesics impinging on a nematic singularity are in fact always driven to the correct direction, $\hat{\boldsymbol{\theta}}^*$.

While Eq. (29) offers some physical insight into the dynamics, a significantly more useful reformulation is possible. The geodesic equation may be expressed directly in terms of the dreibein [Eq. (11b)]. This is natural in this setting, since the dreibein (and not the metric) is fundamental to the formulation of the Dirac field on curved spacetime, Eq. (5). The structure of the quenched gravitational spacetime [Eq. (11c)] allows even further simplification. Relegating the details to Appendix A, we find that the equation *for null geodesics* can be expressed as

$$\dot{x}(t) = \hat{\boldsymbol{\phi}} \cdot \mathbf{v_1}, \tag{30a}$$

$$\dot{y}(t) = \hat{\boldsymbol{\phi}} \cdot \mathbf{v_2}, \tag{30b}$$

$$\dot{\phi}(t) = \hat{\boldsymbol{\phi}} \times \left( \left[ \partial_1 \mathbf{v_1} + \partial_2 \mathbf{v_2} \right] - \frac{1}{\mathbf{v_1} \times \mathbf{v_2}} \left[ [\mathbf{v_1}\partial_1 + \mathbf{v_2}\partial_2][\mathbf{v_1} \times \mathbf{v_2}] \right] \right), \tag{30c}$$

where $\hat{\boldsymbol{\phi}} \equiv [\cos\phi, \sin\phi]^T$ is an auxiliary unit vector that rotates along the geodesic trajectory. In the zero-disorder limit, $\hat{\boldsymbol{\phi}}$ reduces to the velocity unit vector $\hat{\boldsymbol{\theta}}$. The angle $\phi$ expresses the alignment of the tangent vector relative to the spatial components of the dreibein triad. We note that the implementation of the null geodesic constraint reduces our two second-order geodesic equations [Eqs. (19) and (27)] to three first-order equations.

The form of the geodesic equation in Eqs. (30a)–(30c) is useful for numerical simulation; while it is not divergence-free at a curvature singularity, it avoids the singularities in the Christoffel symbols and in the denominator of Eq. (29). Further, the nullity condition $[\Delta = 0$ in Eq. (21)] is implemented automatically by the use of the unit vector $\hat{\boldsymbol{\phi}}$, providing numerical stability. Eqs. (30a)–(30c) also allow easy insight of the geodesic collimation effects of singularities mentioned above; we discuss these next.

### 3.5 Geodesic collimation at nematic singularities

From Eq. (30a)–(30c), we can now better understand geometric features displayed by geodesics in the vicinity of a nematic curvature singularity, what we have been calling *geodesic collimation*. As discussed in Secs. 2.3 and 3.4, all geodesics impinging on a nematic singularity are driven to pass through at the direction defined by the angle $\hat{\boldsymbol{\theta}}^*$ and at the speed $|\mathbf{v}|$. This is depicted for a simple circular geometry in Fig. 4, and for random (quenched disorder) geometry in Fig. 5.

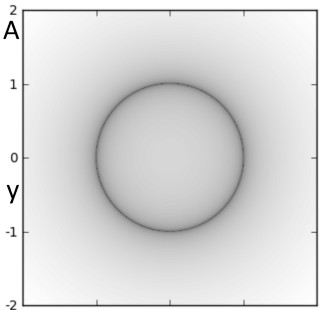
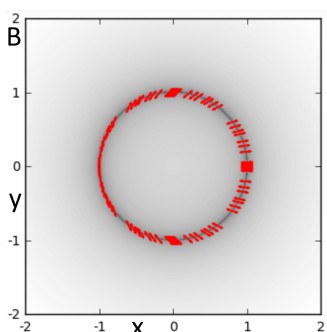
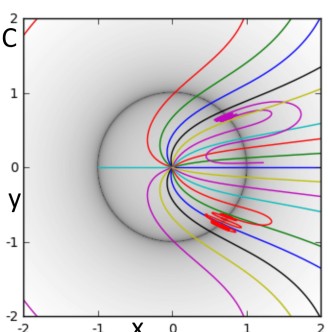

Figure 4: Example of null geodesic collimation. We use a spacetime defined by $v_D = x$ and $v_N = y$ in the purely nematic model of Eqs. (51a) and (51b). A: Heat map of the time-dilation factor $\gamma$ [Eq. (25)], depicting curvature singularities along unit circle. B: Heat map annotated to mark the collimation angles of the singularities. C: Heat map with null geodesic trajectories superimposed. Note that the geodesics pass through the singular manifold at the correct collimation angles. We can also see some geodesics arc back into metastable orbits along the singular manifold.

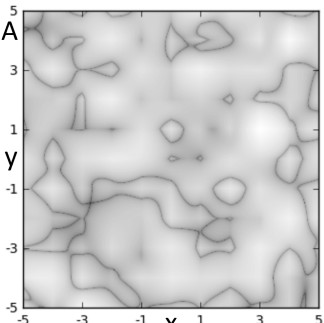
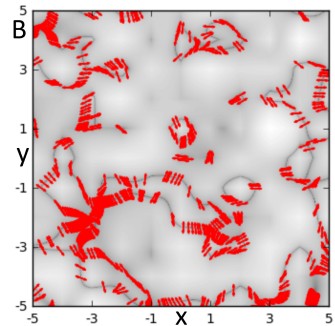
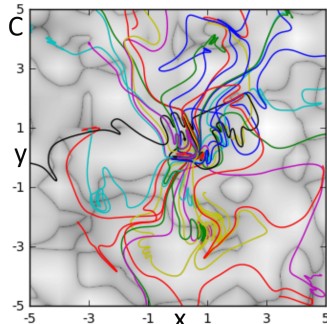

Figure 5: Example of null geodesic collimation in a random quenched gravitational disorder realization. We use the same manifold depicted in Fig. 1. A: Heat map of time-dilation factor $\gamma$ [Eq. (25)], depicting domain walls of singularities. B: Heat map annotated to mark the collimation angles of the singularities. C: Heat map with null geodesic trajectories superimposed. We see both that the geodesics pass through the singular manifold at the correct collimation angles, and that geodesics have a tendency to cross singular manifolds multiple times.

We have seen that at a singularity, the vectors $\mathbf{v}_1$ and $\mathbf{v}_2$ are parallel, and can be parametrized as in Eq. (13). Plugging this into Eqs. (30a) and (30b) then gives $dy/dx = \tan\theta^*$, fixing the direction of the geodesic at the singularity. This is the *collimation angle* $\theta^*$ of the curvature singularity.

Via Eqs. (13), (30a), and (30b), we also have that, at a singularity, the geodesic speed is given by $|\dot{\mathbf{x}}| = \mathbf{v} \cdot \hat{\boldsymbol{\phi}}$. From Eq. (30c), we see that for $\phi$ to have a finite derivative at the singularity, we must have strong driving of $\hat{\boldsymbol{\phi}} \to \hat{\mathbf{v}}$ so that at the singularity, $\hat{\boldsymbol{\phi}}$ is *parallel* to the vectors $\mathbf{v}_1, \mathbf{v}_2$. The geodesic equation thus locks the speed of the geodesic through the singularity to $|\mathbf{v}|$.

## 3.6 Geodesic coincidence at singular points

A geodesic on a Riemannian differentiable manifold is uniquely specified if its position and velocity (tangent vector) at a point are given. Since geodesic collimation at a nematic curvature singularity dictates the velocity of a geodesic at a specific singular point, it would appear that

only a single geodesic may pass through each nematic singular point. This turns out *not* to be the case. A continuum of distinct geodesics may pass through the same singular point at the same time, coinciding in both position and velocity, without contradiction, as shown in Fig. 6. This is possible because our space is only *piece-wise* a Riemannian differentiable manifold, i.e. when restricted to the connected, open sets that are non-singular. As we approach a singularity, the form of the geodesic equation allows it to avoid specifying the value of $\dot{\phi}$ at the singularity, despite the fact that $\{x, y, \phi, \dot{x}, \dot{y}\}$ are completely determined, allowing for distinct geodesics to have the same instantaneous position and velocity (but different values of $\dot{\phi}$). In turn, the value of $\dot{\phi}$ at the singularity uniquely characterizes the geodesic; all higher derivatives of $x, y$ and $\phi$ at the singularity can be computed in terms of $\dot{\phi}$ and the values (and derivatives) of the disorder potentials at the singularity.

To see how geodesic collimation avoids uniquely specifying the geodesics, we linearize Eqs. (30a)–(30c) about a singular point to first order in $t$. This linearization requires the use of a convective derivative; we evaluate potentials along the geodesic and take a total derivative with respect to $t$. We let $\mathbf{v}$ and $\theta^*$ correspond to the singularity as defined by Eq. (13), and (without loss of generality) we take the singularity to be at the origin and the collision to occur at time $t = 0$. We have

$$
\begin{aligned}
\dot{x}(0) &= \cos\theta^* |\mathbf{v}| + \mathcal{O}(t), \\
\dot{y}(0) &= \sin\theta^* |\mathbf{v}| + \mathcal{O}(t), \\
\mathbf{v_1}[\mathbf{x}(0)] &= \cos\theta^* \mathbf{v} + t|\mathbf{v}|(\hat{\boldsymbol{\theta}}^* \cdot \partial)\mathbf{v_1}|_{\mathbf{x}=0} + \mathcal{O}(t), \\
\mathbf{v_2}[\mathbf{x}(0)] &= \sin\theta^* \mathbf{v} + t|\mathbf{v}|(\hat{\boldsymbol{\theta}}^* \cdot \partial)\mathbf{v_2}|_{\mathbf{x}=0} + \mathcal{O}(t), \\
(\mathbf{v_1} \times \mathbf{v_2})[\mathbf{x}(0)] &= t|\mathbf{v}|(\hat{\boldsymbol{\theta}}^* \cdot \partial)(\mathbf{v_1} \times \mathbf{v_2})|_{\mathbf{x}=0} + \mathcal{O}(t^2).
\end{aligned}
\tag{31}
$$

We also expand the unit vector $\hat{\boldsymbol{\phi}}$ about a singularity with collimation angle $\theta$:

$$
\hat{\boldsymbol{\phi}}(t) = \hat{\mathbf{v}} - t\dot{\phi}(0)\hat{\mathbf{v}}^\perp + \mathcal{O}(t^2).
\tag{32}
$$

Plugging these expansions into Eq. (30c), we obtain

$$
\dot{\phi}(t) = \left[\hat{\mathbf{v}} - t\dot{\phi}(0)\hat{\mathbf{v}}^\perp + \mathcal{O}(t^2)\right] \times \left[\hat{\mathbf{v}}\left(\frac{-1}{t} + \mathcal{O}(1)\right) + \mathbf{D} + \mathcal{O}(t)\right],
\tag{33}
$$

where

$$
\mathbf{D} = \left(\partial_1\mathbf{v_1} + \partial_2\mathbf{v_2} - \frac{[(\hat{\boldsymbol{\theta}}^* \cdot \partial\mathbf{v_1})\partial_1 + (\hat{\boldsymbol{\theta}}^* \cdot \partial\mathbf{v_2})\partial_2](\mathbf{v_1} \times \mathbf{v_2})}{(\hat{\boldsymbol{\theta}}^* \cdot \partial)(\mathbf{v_1} \times \mathbf{v_2})}\right)\Bigg|_{\mathbf{x}=0}.
\tag{34}
$$

Carrying out the cross products, we find that $\hat{\mathbf{v}} \times \mathbf{D} = 0$ and that Eq. (30c) simply reduces to

$$
\dot{\phi}(t) = \dot{\phi}(0) + \mathcal{O}(t).
\tag{35}
$$

The $t \to 0$ limit leaves $\dot{\phi}(0)$ *completely undetermined*.

The fact that $\dot{\phi}(0)$ is left undetermined at the singularity opens up the *possibility* that distinct geodesics can share an instantaneous position and velocity. To see that this actually happens, we construct explicit examples from an exactly solvable model—this is done in Sec. 5, but the results are plotted in Fig. 6.

## 3.7 Time-dependent potentials

The form of the geodesic equation in Eqs. (30a)–(30c) provides such a simplification that it is worth checking how this approach fares for time-dependent gravitational potentials. We surprisingly find that this leaves the geodesic equations are *almost* unaltered. With $\mathbf{v_j} \to \mathbf{v_j}[t, \mathbf{x}(t)]$,

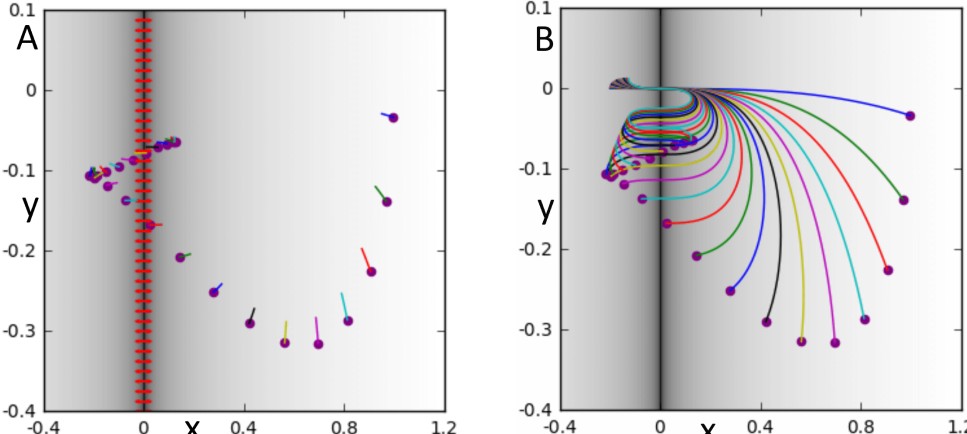

Figure 6: Example demonstrating that *distinct* geodesics can agree in both position and velocity (orientation of the tangent vector) at a singular point. The manifold is the "linear dreibein wall" treated in Sec. 5, and the null geodesics have the closed form solution given by Eqs. (66a) and (66b). A: Several geodesics are launched at $t = 0$ (launch points marked with circles) in the vicinity of a nematic singularity wall along the line $x = 0$, with a horizontal collimation direction (marked with arrows). B: At $t = 1$, all geodesics simultaneously pass through the origin at the correct collimation velocity, as dictated by singularity. All 25 distinct geodesics have the same instantaneous position and velocity at $t = 1$. After traversing the singularity, these mutually diverge (remain distinct) in their subsequent evolution along the manifold.

we still have $\dot{x}^j(t) = \hat{\boldsymbol{\phi}} \cdot \mathbf{v_j}$. The equation for $\phi$ is slightly modified,

$$\dot{\boldsymbol{\phi}}(t) = \hat{\boldsymbol{\phi}} \times \left( \begin{bmatrix} \partial_1 \mathbf{v_1} + \partial_2 \mathbf{v_2} \end{bmatrix} - \frac{1}{\mathbf{v_1} \times \mathbf{v_2}} \Big[ [\mathbf{v_1}\partial_1 + \mathbf{v_2}\partial_2][\mathbf{v_1} \times \mathbf{v_2}] \Big] \\ - \frac{1}{\mathbf{v_1} \times \mathbf{v_2}} \Big[ \mathbf{v_1}(\hat{\boldsymbol{\phi}} \times \partial_0 \mathbf{v_2}) - \mathbf{v_2}(\hat{\boldsymbol{\phi}} \times \partial_0 \mathbf{v_1}) \Big] \right). \tag{36}$$

We see that the time-dependent generalization of the geodesic equation is relatively simple as well, including only a single correction term with a quadratic $\hat{\boldsymbol{\phi}}$ dependence. Further, this form makes it apparent that the singularity-collimation effect survives to the time-dependent generalization. At a singular point, we still have strong driving of $\hat{\boldsymbol{\phi}}$ to $\hat{\mathbf{v}}$. While our focus in this paper is on quenched (static) potentials, this result applies generally to any time-dependent gravitational spacetime expressible in the from given by Eq. (11c), and could have potentially useful applications in future work. We will revisit this in our concluding discussion Sec. 7.

## 4 Isotropic and nematic submodels

In order to qualitatively differentiate the effects of isotropic and nematic fluctuations on the geodesics, we identify two subclasses of quenched gravitation that we will study alongside the general metric in Eq. (11c).

## 4.1 Pure isotropic model

The pure *isotropic* model will be defined by

$$\mathbf{v_1}(\mathbf{x}) = \begin{bmatrix} 1 + v_D(\mathbf{x}) \\ v_N(\mathbf{x}) \end{bmatrix}, \tag{37a}$$

$$\mathbf{v_2}(\mathbf{x}) = \begin{bmatrix} -v_N(\mathbf{x}) \\ 1 + v_D(\mathbf{x}) \end{bmatrix}, \tag{37b}$$

where $v_D, v_N$ are the *diagonal* and *off-diagonal* potentials, respectively. This model has been designed so that $\mathbf{v_1} \cdot \mathbf{v_2} = 0$ and $|\mathbf{v_1}| = |\mathbf{v_2}|$ at every point; it encodes isotropic fluctuations and pseudospin rotations, but does not allow for nematic compression of the Dirac cone. In particular, there will be no nematic singularities—all singular points will host a fully flat local Dirac cone.

In light of the pseudospin invariance of the theory, the dynamics of this model are fully determined by the related model with $\mathbf{v_j} = v(\mathbf{x})\hat{\mathbf{e}}_j$, where $\hat{\mathbf{e}}_j$ is a coordinate unit vector and $v(\mathbf{x})^2 = |\mathbf{v_j}|^2 = \mathbf{v_1} \times \mathbf{v_2} = (1 + v_D)^2 + v_N^2 = 1/\gamma \geq 0$. Singularities occur only at points where $\{v_D = -1, v_N = 0\}$, as depicted in Fig. 7. Plugging the disorder vectors of Eqs. (37a) and (37b) into the constant-interval speed condition [Eq. (29)], we have $\sigma(\theta, \mathbf{x}) = v(\mathbf{x})$ for null geodesics. For the isotropic model, there is no angular dependence of $\sigma$; all geodesics that hit a singularity are stopped, in line with the remarks about isotropic singularities in Sec. 3.5.

In the case of the isotropic model, we find a dramatically simpler form of the metric and

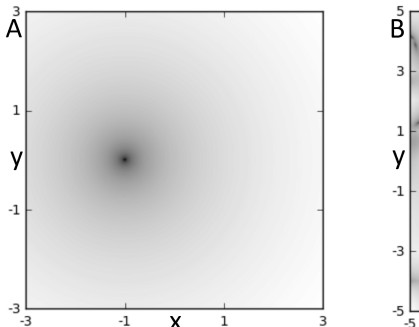 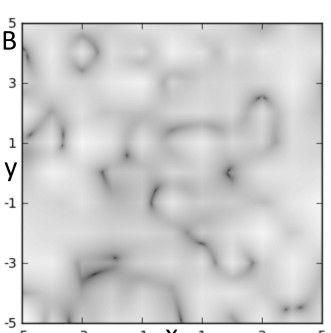 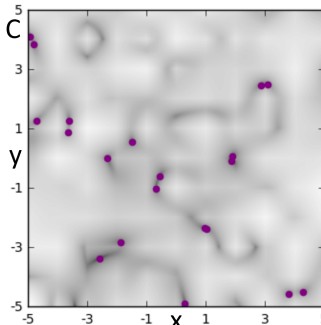

Figure 7: Curvature in the pure isotropic model. A: Heat map of the time-dilation factor $\gamma$ [Eq. (25)] for the "disorder space" isotropic manifold, with $v_D = x$ and $v_N = y$ in Eqs. (37a) and (37b). We note that there is a single isolated curvature singularity. B: A heat map of the time-dilation factor for a random realization of pure isotropic quenched gravity. C: The heat map in B annotated to mark the locations of isotropic singularities.

Christoffel symbols,

$$g_{\mu\nu}(\mathbf{x}) \rightarrow \begin{bmatrix} -v(\mathbf{x})^2 & 0 & 0 \\ 0 & 1 & 0 \\ 0 & 0 & 1 \end{bmatrix}, \tag{38}$$

$$\Gamma^0_{\mu\nu}(\mathbf{x}) \rightarrow \begin{bmatrix} 0 & \partial_1 & \partial_2 \\ \partial_1 & 0 & 0 \\ \partial_2 & 0 & 0 \end{bmatrix} \log[v(\mathbf{x})], \tag{39}$$

$$\Gamma^1_{\mu\nu}(\mathbf{x}) \rightarrow \begin{bmatrix} v(\mathbf{x})\partial_1 v(\mathbf{x}) & 0 & 0 \\ 0 & 0 & 0 \\ 0 & 0 & 0 \end{bmatrix}, \tag{40}$$

$$\Gamma^2_{\mu\nu}(\mathbf{x}) \rightarrow \begin{bmatrix} v(\mathbf{x})\partial_2 v(\mathbf{x}) & 0 & 0 \\ 0 & 0 & 0 \\ 0 & 0 & 0 \end{bmatrix}. \tag{41}$$

From Eq. (24), we have $dt/ds = v(\mathbf{x})^{-2}$ (setting the constant $E/m = 1$). The spatial geodesic equations are

$$\frac{d^2\mathbf{x}}{ds^2} = \boldsymbol{\nabla}\left[\frac{1}{2v(\mathbf{x})^2}\right] = \boldsymbol{\nabla}\left[\frac{\gamma(\mathbf{x})}{2}\right]. \tag{42a}$$

With respect to the affine parameter $s$, these equations represent a non-relativistic classical-Hamiltonian system with the effective potential

$$U(\mathbf{x}) = -\frac{1}{2}\gamma(\mathbf{x}). \tag{43}$$

Conservation of energy for the Hamiltonian system takes the form

$$\frac{1}{2}\left|\frac{d\mathbf{x}}{ds}\right|^2 = \mathcal{E} + \frac{1}{2}\gamma(\mathbf{x}). \tag{44}$$

Above, $\mathcal{E}$ determines both the Hamiltonian energy of the classical system and the length of the spacetime interval:

$$\Delta(s) = g_{\mu\nu}[\mathbf{x}(s)]\frac{dx^\mu}{ds}\frac{dx^\nu}{ds} = 2\mathcal{E}. \tag{45}$$

A null geodesic is a trajectory with $\mathcal{E} = 0$.

In this picture, the factor $\gamma = 1/v^2(x)$ [Eq. (25)] enters acts as a potential energy $U(\mathbf{x})$, and singularities are infinitely deep potential wells. It is worth asking how the capture of geodesics by isotropic singularities is compatible with Eq. (44); conservation of the energy $\mathcal{E}$ would seem to prevent geodesics from terminating at an isotropic singularity. The key point is that energy conservation only holds when geodesics are expressed in terms of the affine parameterization. Indeed, solving the geodesic equation in terms of the affine parameter, one finds trajectories that pass right through singularities. However, because $U \rightarrow -\infty$ corresponds to infinitely strong time dilation [Eq. (25)], when the geodesic is reparametrized in terms of the global time coordinate, $t$, the spatial coordinates $\mathbf{x}(t)$ instead slow upon approaching the singular point, and collide with it only as $t \rightarrow \infty$. The actual point of collision with the singularity is time-dilated to the infinite future. When we instead first reparametrize the geodesic equation by global time and then solve directly for geodesics in terms of $t$, the loss of conservation of energy due to time dilation can be attributed to the additional (non-Christoffel) "friction terms"

appearing on the right-hand side of Eq. (27) that arise due to the time reparametrization. In Sec. 5.1, we consider a highly symmetric geometry with an isotropic singularity at the origin that can be solved exactly. In that case, we will see explicitly how geodesic capture occurs in both frames of reference.

The analogue of Eqs. (30a)–(30c) for the isotropic model is also much simpler,

$$\dot{\mathbf{x}} = v(\mathbf{x})\hat{\boldsymbol{\phi}} \tag{46a}$$

$$\dot{\phi} = [\boldsymbol{\nabla} v(\mathbf{x})] \times \hat{\boldsymbol{\phi}}. \tag{46b}$$

In this case, we see that $\hat{\boldsymbol{\phi}}$ simply gives the velocity direction of the geodesic, and $v(\mathbf{x})$ is the speed. The velocity vector rotates when it is not aligned with the gradient of $v(\mathbf{x})$.

Finally, in the case of the isotropic model the Ricci scalar curvature is simple enough to state:

$$R(\mathbf{x}) = -\frac{2\nabla^2 v}{v}. \tag{47}$$

The fully quantum-mechanical formulation of the pure isotropic model can be re-written so that its time-dependent wave function is determined by the solution of an auxiliary Hermetian differential equation. To see this, note that pseudospin invariance asserts that the dynamics of the general isotropic model can be studied by the action

$$\mathcal{S} = i \int dt \int d^2\mathbf{x}\, \bar{\psi}(t,\mathbf{x}) \begin{pmatrix} \partial_t + v(\mathbf{x})\hat{\sigma}^1\partial_1 + v(\mathbf{x})\hat{\sigma}^2\partial_2 \\ + \frac{1}{2}[\partial_1 v(\mathbf{x})]\hat{\sigma}^1 + \frac{1}{2}[\partial_2 v(\mathbf{x})]\hat{\sigma}^2 \end{pmatrix} \psi(t,\mathbf{x}), \tag{48}$$

from which we can extract the Schrödinger equation. We can deal with the spin connection terms [the second line in Eq. (48)] by introducing $\tilde{\psi}(t,\mathbf{x}) = \sqrt{v(\mathbf{x})}\,\psi(t,\mathbf{x})$, which satisfies

$$\{i\partial_t + v(\mathbf{x})[\hat{\sigma}^1 i\partial_1 + \hat{\sigma}^2 i\partial_2]\}\,\tilde{\psi}(t,\mathbf{r}) = 0. \tag{49}$$

Dividing by $v(\mathbf{x})$, we see the energy eigenstates are determined by the "spatially-random energy" Dirac equation

$$-i\left[\hat{\sigma}^1\partial_1 + \hat{\sigma}^2\partial_2\right]\tilde{\psi}_E(\mathbf{x}) = \frac{E}{v(\mathbf{x})}\tilde{\psi}_E(\mathbf{x}). \tag{50}$$

## 4.2 Pure nematic model

The pure *nematic* model will be defined by

$$\mathbf{v_1}(\mathbf{x}) = \begin{bmatrix} 1 + v_D(\mathbf{x}) \\ v_N(\mathbf{x}) \end{bmatrix}, \tag{51a}$$

$$\mathbf{v_2}(\mathbf{x}) = \begin{bmatrix} v_N(\mathbf{x}) \\ 1 - v_D(\mathbf{x}) \end{bmatrix}, \tag{51b}$$

where $v_D, v_N$ are the *diagonal* and *non-diagonal* potentials, respectively.

This model has been designed to encode nematic fluctuations, but does not allow for isotropic compression of the Dirac lightcone. In particular, there can be no isotropic singularities. We note that $\mathbf{v_1} \times \mathbf{v_2} = 1 - (v_D^2 + v_N^2) = 1/\gamma$, so that the curvature singularities fall along the unit circle in $\{v_D, v_N\}$-space, as depicted in Fig. 8. Plugging the disorder vectors of Eqs. (51a)–(51b) into the constant-interval speed condition [Eq. (29)], for null geodesics we have

$$\sigma(\theta, \mathbf{x}) = \frac{|1 - (v_D^2 + v_N^2)|}{\sqrt{[v_D - \cos(2\theta)]^2 + [v_N - \sin(2\theta)]^2}}. \tag{52}$$

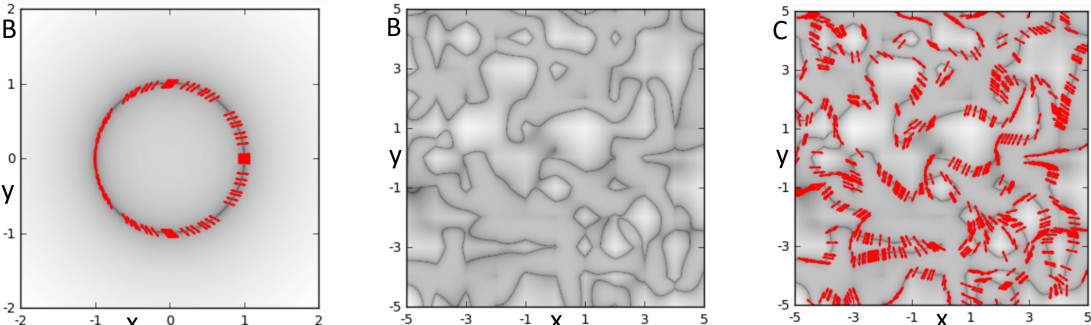

Figure 8: Curvature in the pure nematic model. A: Heat map of the time-dilation factor $\gamma$ [Eq. (25)] for the "disorder space" nematic manifold, with $v_D = x$ and $v_N = y$ in Eqs. (51a) and (51b). We note the singularities lie along the unit circle, with the collimation angle marked in red. Numerically computed null geodesics are displayed in Fig. 4 for this geometry. B: A heat map of the time-dilation factor for a random realization of pure nematic QGD. C: The heat map in B annotated to mark the collimation angles of the nematic singularities.

We see that for the nematic model, there is an angular dependence of $\sigma$ and the collimation angle of a singularity is closely tied to the geometry of the unit circle in $\{v_D, v_N\}$-space.

Unlike the pure isotropic model, the pure nematic model neither yields a significantly simplified form of the geodesic equation nor a partial solution to the quantum problem analogous to Eq. (50). It is related to the $T\bar{T}$ deformation of 2D quantum field theories [17, 34]. The Ricci scalar curvature is extremely unwieldy and not particularly useful.

# 5 Solvable manifolds with curvature singularities

In this section we present several "toy models" of quenched gravitation, that is, highly symmetric realizations of the velocity potentials $\mathbf{v}_{1,2}(\mathbf{x})$ in Eq. (11c) that allow (full or partial) analytical solution to the geodesic equation. We have several motivations here. Firstly, we observe in numerical solutions that geodesics are often captured by isotropic singularities or drawn into meta-stable gravitationally bound orbits along nematic singularity walls. We would like to understand these phenomena through the lens of some exactly solvable models. In particular, we want closed form solutions that shed light on the nature of bound state orbits and on the asymptotic approach to a singular point. Further, we can use analytical solutions to benchmark our numerical solver.

## 5.1 Isotropic power-law model

We observe in numerical results that isotropic singularities tend to be highly attractive, and that geodesics can be captured by these. One may ask if this is an artifact of the numerical solver or if the geodesics truly asymptote to the singularities. Here we study a family of integrable examples of the purely isotropic model to see how the geodesics approach such a singularity.

We will consider the pure isotropic model with $v(r) = r^\alpha$ for $\alpha > 0$ (in the notation of Sec. 4.1). Adapting to polar coordinates $(r, \theta)$ in the plane, one finds that the geodesic

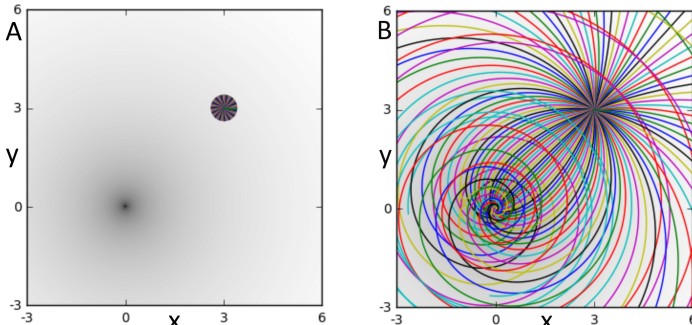

Figure 9: Null geodesic trajectories for the isotropic power-law model with $\alpha = 1$. A: Geodesics released at $(3,3)$ with various launch angles, $\phi_0$. B: Long-term geodesic dynamics, in agreement with Eqs. (54a) and (54b). With $\mathbf{x}_0 = (3,3)$, $\theta_0 = \pi/4$, so we have captured orbits for $\phi_0 \in [3\pi/4, 7\pi/4]$ and escaping orbits for $\phi_0 \in [-\pi/4, 3\pi/4]$.

equations read

$$\dot{r}(t) = r^\alpha \cos\left[\phi(t) - \theta(t)\right], \tag{53a}$$

$$\dot{\theta}(t) = r^{\alpha-1} \sin\left[\phi(t) - \theta(t)\right], \tag{53b}$$

$$\dot{\phi}(t) = \alpha r^{\alpha-1} \sin\left[\phi(t) - \theta(t)\right]. \tag{53c}$$

We stress that here $\theta$ denotes the positional angle in the plane, *not* the orientation of the tangent (velocity) vector, as employed elsewhere in this paper. The Eqs. (53) are easy to solve; integrating gives $\phi(t) = \alpha\theta(t) + c_0$ for some initial-value constant $c_0$.

In the *marginal* $\alpha = 1$ case (see below), the null geodesics are given by

$$r(t) = r_0 e^{\cos(c_0)t} \tag{54a}$$

$$\theta(t) = \sin(c_0)t + \theta_0. \tag{54b}$$

These geodesics rotate with a constant angular velocity and they either decay towards the origin or explode outwards exponentially. Geodesics that start out heading towards the singularity are always captured and those that start out heading away always escape. This is depicted in Fig. 9.

For a generic potential that can be Taylor-expanded in $r$ about an isotropic singularity, the $\alpha = 1$ case considered above captures the lowest-order term in the expansion. This result provides intuition that a geodesic that enters a sufficiently small neighborhood of an isotropic singularity heading towards it will be asymptotically captured.

We can also treat the general case. For $\alpha \neq 1$, define the function $\beta(t) \equiv \phi(t) - \theta(t) = (\alpha - 1)\theta(t) + c_0$. With this, the geodesic equations then have a first integral of the form

$$\frac{\sin[\beta(t)]}{\sin\beta_0} = \left[\frac{r(t)}{r_0}\right]^{\alpha-1}. \tag{55}$$

We can see from this formula that for $\alpha > 1$, all geodesics with $\sin\beta_0 \neq 0$ are bound for these manifolds. This allows to solve for $\beta(t)$ and $r(t)$:

$$\cot[\beta(t)] = (1-\alpha)\frac{r_0^{\alpha-1}}{\sin\beta_0}t + \cot\beta_0, \tag{56a}$$

$$r(t) = \left\{\frac{\sin\beta_0}{r_0^{\alpha-1}}\sqrt{1 + \left[(1-\alpha)\frac{r_0^{\alpha-1}}{\sin\beta_0}t + \cot\beta_0\right]^2} + c_1\right\}^{-1/(\alpha-1)}, \tag{56b}$$

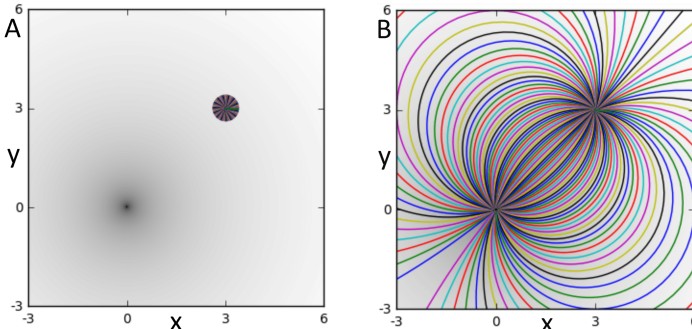

Figure 10: Geodesic trajectories for the isotropic power-law model with $\alpha = 2$. A: Geodesics released at $(3,3)$ with various angles. B: Long-term geodesic dynamics, in agreement with Eqs. (56a) and (56b).

where $c_1$ is a constant determined by the $t \to 0$ limit. We see from Eq. (56) that $r(t)$ approaches zero in an $\alpha$-dependent power law for $\alpha > 1$, and that the asymptotic angle of approach to the singularity is only dependent on $\theta_0, \phi_0$. This is depicted in Fig. 10.

This model allows us to see explicitly how time-dilation allows isotropic singularities to capture geodesics despite conservatino of energy. The pure isotropic model is described by a Hamiltonian system in terms of the affine parameter [Eqs. (42a) and (43)]. For the isotropic power-law model $\gamma(r) = 1/r^{2\alpha}$, we have the conservation law

$$\frac{1}{2}\left(\frac{dr}{ds}\right)^2 + \frac{l^2}{2r^2} - \frac{1}{2r^{2\alpha}} = \mathcal{E}, \tag{57}$$

where $l = r^2(d\theta/ds)$ is the angular momentum. For $\alpha < 1$, the effective radial potential diverges to $+\infty$ as $r \to 0$, and no trajectories cross the singularity. For $\alpha > 1$, the effective potential diverges to $-\infty$, and all trajectories cross the singularity. This all agrees with the closed form solution for null geodesics, Eqs. (56a) and (56b). While Eq. (57) implies that the kinetic term diverges when a geodesic crosses a singularity, this is only with respect to the affine parametrization; time dilation effects overwhelm that divergence and when parameterized in terms of global coordinate time $t$, the geodesics slow and asymptote to the singularity.

The case $\alpha = 1$ is marginal, and only trajectories with $l^2 > 1$ are blocked from the singularity by the centrifugal barrier. We can see how this works from the solution in Eq. (54). Naively calculating $l^2$ from these would give a non-constant angular momentum, because these solutions are given in terms of the global coordinate time. Re-expressing Eq. (54) in terms of the affine parametrization, we have

$$\begin{aligned}\frac{dr}{ds} &= \frac{1}{r}\cos(c_0), \\ \frac{d\theta}{ds} &= \frac{1}{r^2}\sin(c_0).\end{aligned} \tag{58}$$

We see that $l^2 = \sin(c_0)^2 \le 1$, so that in this case, none of our geodesics are centrifugally prevented from crossing the singularity. As a result, the solution in Eq. (54) asymptotes to $r = 0$ in either the infinite future or past. The exception is the $l^2 = 1$ orbit, with $\cos(c_0) = 0$, which orbits at fixed radius.

## 5.2 xy-factored model

In this section, we present a class of 2D toy models that is solvable due to a "factorization" into independent 1D structures. It provides a class of example manifolds on which both nematic

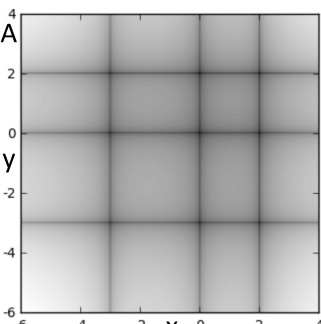
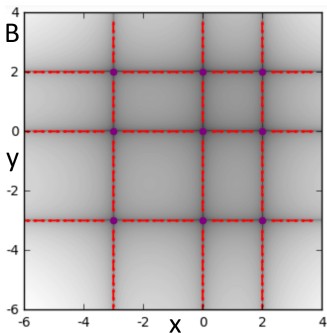
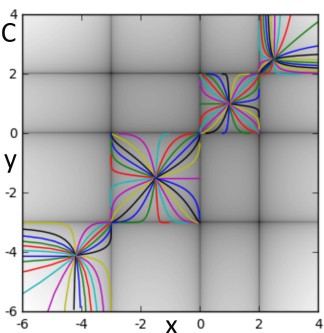

Figure 11: Geodesic trajectories in the xy-factored model. We use a manifold with $v_1^1(x) = p(x)$, $v_2^2(y) = p(y)$ with $p(x) = x^3 + x^2 - 6x$, creating curvature singularities along the lines $\{x = -3,\ x = 0,\ x = 2,\ y = -3,\ y = 0,\ y = 2\}$. A: Heat map of the time-dilation factor $\gamma$ [Eq. (25)], depicting curvature singularities along vertical and horizontal walls. B: Heat map annotated to mark the collimation angles of the nematic singularities and the locations of isotropic singularities. C: Heat map with geodesic trajectories superimposed. Note that the geodesics are trapped in the box in which they start and asymptote towards isotropic singularities at the corners.

and isotropic singularities attract geodesics; all geodesics asymptote towards nematic singularity walls, and run along these walls until finally being captured by an isotropic singularity.

We are interested in the class of models with $\mathbf{v_1} \cdot \mathbf{v_2} = 0$ and $\partial_2|\mathbf{v_1}| = \partial_1|\mathbf{v_2}| = 0$ everywhere. In light of the pseudospin invariance of the geodesic dynamics, we may reduce to the model defined by the disorder vectors $\mathbf{v_1} = v_1^1(x)\hat{\mathbf{e}}_1$ and $\mathbf{v_2} = v_2^2(y)\hat{\mathbf{e}}_2$. We then have $|\mathbf{v_1} \times \mathbf{v_2}| = v_1^1(x)v_2^2(y)$, so that singularities fall along the vertical and horizontal lines defined by the zeros of $\{v_1^1, v_2^2\}$. Let $\{x_j\}$ and $\{y_j\}$ denote the zeros of $v_1^1$ and $v_2^2$, respectively; we note that they partition the plane into rectangular boxes, $B_{ij} = (x_i, x_{i+1}) \times (y_j, y_{j+1})$ (see Fig. 11), separated by walls of nematic singularities and with isotropic singularities at the corners.

We solve for the geodesic dynamics. In this setting, the geodesic equations (30a)–(30c) reduce to

$$\dot{x}(t) = v_1^1(x)\cos\phi\,, \tag{59a}$$

$$\dot{y}(t) = v_2^2(y)\sin\phi\,, \tag{59b}$$

$$\dot{\phi}(t) = 0\,. \tag{59c}$$

That $\phi$ is constant along all geodesics allows the geodesics to be written in closed form. We define the functions

$$F_j(x) = \int_{(x_j + x_{j+1})/2}^{x} \frac{dz}{v_1^1(z)}\,, \tag{60a}$$

$$G_j(y) = \int_{(y_j + y_{j+1})/2}^{y} \frac{dz}{v_2^2(z)}\,. \tag{60b}$$

The mapping $(x, y) \longleftrightarrow \big(F_i(x), G_j(y)\big)$ provides a diffeomorphism between the box $B_{ij}$ and the plane. Geodesics in the box $B_{ij}$ are described by

$$x(t) = F_i^{-1}[t\cos\phi_0 + F_i(x_0)]\,, \tag{61a}$$

$$y(t) = G_j^{-1}[t\sin\phi_0 + G_j(y_0)]\,, \tag{61b}$$

which we see implies that geodesics never escape the $B_{ij}$ regions that they originate in; they asymptotically approach the isotropic singularities in the corners of the $B_{ij}$ regions, riding along the nematic singularity walls. We plot an example in Fig. 11

Here we have a concrete example of null geodesics asymptotically approaching both nematic singularity walls and isotropic singularities, with capture by isotropic singularities, and a new perspective on the interactions between nematic and isotropic singularities in models where both are allowed. It also offers perspective on what happens when the collimation angle of a nematic singularity is fixed to be parallel to the singularity manifold: such singularities seem to be impossible for geodesics to cross.

## 5.3 Dreibein wall model

We next consider a model with a dreibein wall. The goal is to understand gravitationally bound orbits of geodesics that cross a nematic singularity wall many times, as observed often in numerical solutions (e.g., Fig. 3). We define the model by the disorder vectors $\mathbf{v_1} = \hat{\mathbf{e}}_1$ and $\mathbf{v_2} = m(x)\,\hat{\mathbf{e}}_2$. We choose $m(x)$ such that $m(0) = 0$ to place the nematic singularity wall along the $y$-axis. The model has geodesic collimation angle $\theta^* = 0$ (perpendicular to the dreibein wall); walls with other collimation angles are easily constructed, but exhibit qualitatively similar physics.

The null geodesic equations (30a)–(30c) reduce to

$$\dot{x}(t) = \cos[\phi(t)], \tag{62a}$$

$$\dot{y}(t) = m[x(t)]\sin[\phi(t)], \tag{62b}$$

$$\dot{\phi}(t) = \frac{m'[x(t)]}{m[x(t)]}\sin[\phi(t)], \tag{62c}$$

which admit a general first integral of the form

$$\frac{\sin[\phi(t)]}{\sin\phi_0} = \frac{m[x(t)]}{m_0}. \tag{63}$$

We note that along a trajectory we must have $|m[x(t)]| < |m_0/\sin\phi_0|$. This condition will provide a very simple way to understand and compute trapping horizons for gravitationally bound orbits. In particular, we can see that for unbounded $m(x)$, *all* geodesic trajectories with $\sin\phi_0 \neq 0$ are bound.

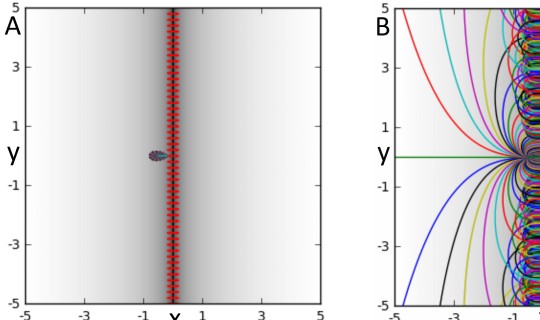

Figure 12: Null geodesic trajectories for the linear dreibein wall, with $m(x) = x$. A: Geodesics launched near the wall of singularities. The singular manifold has been labeled with collimation arrows. B: The long-time geodesic dynamics. Note that all geodesics are in permanent bound states along the singular wall, see Eq. (66). All crossings of the singular wall occur at the collimation angle $\theta^* = 0$ (or $\pi$).

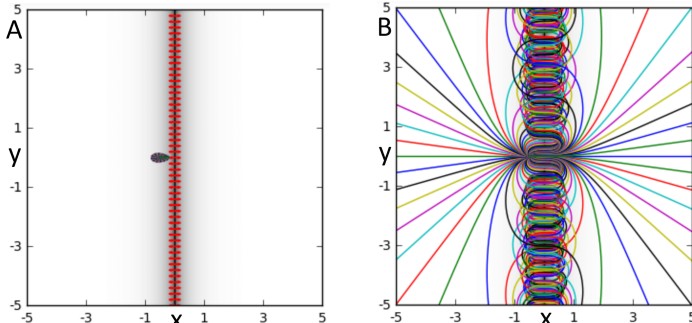

Figure 13: Null geodesic trajectories for the dreibein wall, with $m(x) = \tanh(x)$. A: Geodesics launched near the wall of singularities. The singular manifold has been labeled with collimation arrows. B: The long-time geodesic dynamics. Note that some geodesics are in permanent bound states along the singular wall, while others escape to infinity. The separation between bound and free orbits is determined by Eq. (67). All crossings of the singular wall occur at the collimation angle $\theta^* = 0$ (or $\pi$).

We may use Eq. (63) to reformulate the geodesic equations as a first order system in the position variables:

$$\dot{x}(t) = \pm\sqrt{1 - \left[\sin\phi_0\frac{m(x)}{m_0}\right]^2}, \tag{64a}$$

$$\dot{y}(t) = \sin\phi_0\frac{m[x(t)]^2}{m_0}. \tag{64b}$$

We can extract the behavior of null geodesics near a trapping horizon by linearizing Eq. (64) around a point $x_*$ such that $m(x_*) = |m_0/\sin\phi_0|$. Assuming (without loss of generality) that $m'(x_*) > 0$ gives

$$x(t_*) = x_* - \left|\frac{\sin\phi_0}{2m_0}\right|m'(x_*)t_*^2, \tag{65}$$

where $t_*$ is a shifted time coordinate defined so that the impact with the horizon occurs at $t_* = 0$. Importantly, we see that the trapping horizon is *not* an asymptote, but a turning point that sends the geodesic back the other way in finite time. This gives an example by which we can understand gravitational bound-state orbits of geodesics along singularity walls.

We consider the case of a linear dreibein wall, with $m(x) = x$. In this case, we can directly integrate Eq. (64) to obtain

$$x(t) = \frac{x_0}{\sin\phi_0}\sin\left[\phi_0 + \frac{\sin\phi_0}{x_0}t\right], \tag{66a}$$

$$y(t) = y_0 + \frac{x_0}{2\sin\phi_0}\left[t + \frac{x_0}{2\sin\phi_0}\left(\sin[2\phi_0] - \sin\left[2\phi_0 + \frac{2\sin\phi_0}{x_0}t\right]\right)\right]. \tag{66b}$$

All geodesics (with $\sin\phi_0 \neq 0$) are bound states, oscillating back and forth across the singularity wall while drifting along it, as shown in Fig. 12. This is a particularly nice example for constructing distinct geodesics that have equal instantaneous positions and velocities at a singularity crossing, and is used to generate the example given in Fig. (6).

We also consider the case $m(x) = \tanh(x)$. This model still has a nematic singularity wall along the $y$-axis, but since $m(x)$ is bounded, not all trajectories will be bound states. In fact,

the condition for a bound-state null trajectory is

$$\left| \frac{\tan(\theta_0)}{\tanh(x_0)} \right| \frac{1}{\sqrt{\tanh(x_0)^2 + \tan(\theta_0)^2}} \geq 1 \,, \tag{67}$$

where $\theta_0$ is the initial launch angle of the geodesic, initially located at $x_0$. We see that the initial launch angle and initial distance from the singularity wall together determine if a geodesic is asymptotically bound or free. We plot geodesic trajectories in Fig. 13.

Again, these toy model solutions add perspective to the results of numerical simulation. They give an analytical understanding of the ability of nematic singularity walls to trap geodesics into oscillatory, gravitationally bound orbits. We expect that the linear profile represents the lowest-order approximation to the curvature profile in the vicinity of a generic nematic singularity wall. These models are designed so that the collimation angle of the geodesics is orthogonal to the singularity manifold at all points, and we see that these orbits are fully stable.

### 5.4 Circular nematic model

The previous section on dreibein wall geometries shows that nematic singularity walls can host states of permanently bound geodesics. An interesting question is whether this is a feature unique to infinite singular walls. In this section, we construct a class of geometries hosting gravitationally bound geodesics along a finite, closed contour.

Our model will be a rotationally symmetric version of the purely nematic model defined by Eqs. (51a) and (51b), with

$$v_D = \rho(r)\cos(2\theta), \tag{68a}$$
$$v_N = \rho(r)\sin(2\theta). \tag{68b}$$

Here $\theta$ denotes the positional polar angle in the plane, *not* the orientation of the tangent (velocity) vector, as employed elsewhere in this paper. The metric [Eq. (11c)], converted to polar spacetime coordinates $(t, r, \theta)$, takes the form

$$g_{\mu\nu} = \frac{1}{1-\rho^2(r)} \begin{bmatrix} -[1-\rho^2(r)]^2 & 0 & 0 \\ 0 & [1-\rho(r)]^2 & 0 \\ 0 & 0 & r^2[1+\rho(r)]^2 \end{bmatrix}, \tag{69}$$

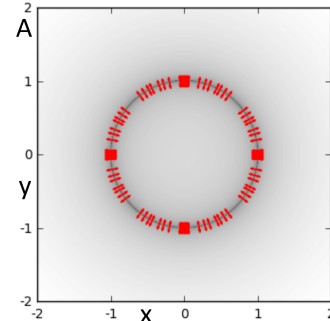
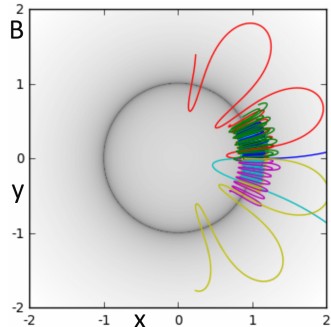

Figure 14: Null geodesic trajectories for the rotationally symmetric nematic model defined by Eq. (68), with $\rho(r) = r$. A: The singular manifold $r^* = 1$, labeled with collimation arrows. B: Geodesic dynamics. Note that many geodesics are in bound states along the singular wall, and all wall crossings occur at the correct collimation angle. The condition for bound-state null geodesics is $e < 1$, where the eccentricity $e$ is defined in Eq. (74).

which becomes singular at $\rho(r) = 1$. The metric is invariant under rotations $\theta \to \theta + \theta_0$; the angular $\theta$-direction is flattened along each radial contour with $\rho(r) = 1$, corresponding to a nematic curvature singularity. The Ricci scalar curvature is given by

$$R(r) = -\frac{2}{[1-\rho(r)]^3} \left\{ \frac{[3-\rho(r)][1-\rho(r)]\rho'(r)}{r} + \frac{[\rho'(r)]^2}{1+\rho(r)} + [1-\rho(r)]^2 \rho''(r) \right\}, \quad (70)$$

where $\rho'(r) = d\rho/dr$.

Converting the geodesic equations (30a)–(30c) to polar coordinates, we find

$$\dot{r} = [1+\rho(r)]\cos[\theta-\phi], \tag{71a}$$

$$\dot{\theta} = -\left[\frac{1-\rho(r)}{r}\right]\sin[\theta-\phi], \tag{71b}$$

$$\dot{\phi} = \left[\frac{1+\rho(r)}{1-\rho(r)}\rho'(r) + 2\frac{\rho(r)}{r}\right]\sin[\theta-\phi]. \tag{71c}$$

Defining $\beta \equiv \theta - \phi$, as before, these equations are separable. The orbit equation relating $r$ to $\beta$ can be directly integrated, yielding

$$\left(\frac{\sin\beta}{\sin\beta_0}\right) = \frac{r_0}{r}\left[\frac{1-\rho(r)}{1-\rho(r_0)}\right]. \tag{72}$$

For the case $\rho(r) = r$, which hosts a singularity wall at $r^* = 1$, the orbit equation becomes

$$r(\beta) = \left[1 + \frac{1-r_0}{r_0}\left(\frac{\sin\beta}{\sin\beta_0}\right)\right]^{-1}. \tag{73}$$

This is a conic section orbit in the $(r,\beta)$ plane with semilatus rectum $\alpha = 1$ and eccentricity

$$e = \frac{|1-r_0|}{r_0\,|\sin(\theta_0-\phi_0)|}. \tag{74}$$

The condition for a bound orbit is $e < 1$.

For the class of power-law models, $\rho(r) = r^\alpha$, Eq. (72) implies that

$$\left|\frac{1-r^\alpha}{r}\right| \le M_0 \equiv \left|\frac{1}{\sin\beta_0}\frac{1-r_0^\alpha}{r_0}\right|, \tag{75}$$

showing that geodesics with $\sin\beta_0 \neq 0$ are forbidden from reaching the origin by a centrifugal barrier and that for $\alpha > 1$, these geodesics are also bounded. This class of models show that closed, finite singular manifolds can host permanent bound states.

## 6 Quantum treatments of toy models

Our work's focus is the geometry of geodesics as a route towards understanding the semiclassical effects of QGD on 2D Dirac fermions. However, it is important to raise the question of just what effects the geodesic flow field of the underlying manifold have on the fully quantum solution to the QGD problem. Some general statements can be made: individual spinor components of the solution of the curved-space Dirac equation are themselves also solutions to a second-order wave equation on the same manifold. (This is the curved-space analog of "squaring" a Dirac equation to a Kline-Gordon equation.) As mentioned in the Introduction, it is well known that the null geodesics of the underlying manifold constitute the *bicharacteristics*

for both the curved-space Dirac equation and the associated wave equations. In the theory of curved space wave equations, discontinuities and shock fronts propagate along bicharacteristics and a study of bicharacteristics is generally equated to a "geometric optics" approximation to the wave equation [29, 30]. While this justifies our focus on geodesics as a "semiclassical" approach in general, we can strengthen the argument by comparing the solutions to the geodesic equation and the fully-quantum Dirac equation in geometries where both may be obtained.

In this section, we revisit the "toy models" introduced in Sec. 5 from the fully quantum picture. We present results on exact energy states in these models that are generally in harmony with the geodesic picture from Sec. 5. Since we expect the geodesics to relate to quantum *dynamics*, which for the most part we are unable to treat analytically, direct comparison of the two pictures is difficult. However, in two exceptions – the "isotropic power law model" in the special case $\alpha = 2$ and the "xy-factored orthogonal model" – we can get analytical quantum dynamics results and find strong agreement with the geodesic picture. We discuss these qualitative similarities in our concluding remarks in Sec. (7).

## 6.1 Isotropic power law model

We first look at the quantum mechanics of the isotropic power-law model in Sec. 5.1. We have seen [Eq. (50)] that the problem of generic *isotropic* fluctuations reduces to solving the equation

$$v(\mathbf{r})\big[\hat{\sigma}^1\partial_1 + \sigma^2\partial_2\big]\tilde{\psi}(\mathbf{r}) = iE\tilde{\psi}(\mathbf{r}).\tag{76}$$

For radially symmetric $v(r)$, we introduce the angular momentum eigenstates:

$$\tilde{\psi}_{l,E}(r,\theta) = e^{il\theta}\begin{bmatrix}1 & 0\\ 0 & e^{i\theta}\end{bmatrix}\tilde{\psi}_{l,E}(r).\tag{77}$$

The surprisingly elegant solutions to the time-independent Schrödinger equation for the isotropic power-law model are then given by [see Appendix. D.1]:

$$\psi_{l,E}^{(\alpha)}(r,\theta) = \left[\frac{1}{4\pi}\left|\frac{E}{1-\alpha}\right|\right]^{1/2}\frac{e^{il\theta}}{r^\alpha}\begin{cases}\begin{bmatrix}\mathcal{J}_m(Ez)\\ -ie^{i\theta}\mathcal{J}_{m-1}(Ez)\end{bmatrix}, & l\ge 0,\\[12pt] \begin{bmatrix}\mathcal{J}_{-m}(Ez)\\ ie^{i\theta}\mathcal{J}_{1-m}(Ez)\end{bmatrix}, & l\le -1,\end{cases}\tag{78}$$

where $E, l$ are energy and angular momentum quantum numbers, $\mathcal{J}_m$ denotes a Bessel function of first kind of order $m$, with

$$m = \frac{l+\alpha/2}{\alpha-1}.\tag{79}$$

We've also introduced the notation

$$z = \frac{r^{1-\alpha}}{1-\alpha}.\tag{80}$$

We note that the above solution breaks down in the $\alpha \to 1$ limit, where we instead find non-normalizable power-law solutions to the Dirac equation.

For $\alpha > 1$, the wavefunctions [Eq. (78)] are physically very strange. Near the singular point at the origin, the probability density oscillates arbitrarily rapidly between nodes and antinodes and has a diverging envelope.

It is interesting that while all geodesics for these models ($\alpha > 1$) are bound [Eq. (55)], instead of a discrete bound-state energy spectrum we find a continuum of scattering states. Despite this, we can show that for the $\alpha = 2$ model, all quantum density asymptotically accumulates at the origin, mimicking the behavior of the geodesics. We give a proof of this in Appendix E.

## 6.2 xy-factored model

We revisit the "xy-factored orthogonal model" from Sec. (5.2) from the quantum perspective.
The time-dependent curved-space Dirac equation reads

$$\left[\partial_t + v_1^1(x)\hat{\sigma}_1\partial_x + v_2^2(y)\hat{\sigma}_2\partial_y + \frac{1}{2}\left(\dot{v}_1^1(x)\hat{\sigma}_1 + \dot{v}_2^2(y)\hat{\sigma}_2\right)\right]\psi(t,x,y) = 0\,. \tag{81}$$

We can make progress by using the functions $F_j, G_j$ defined in Eqs. (60a,60b) and the same "prefactor trick" that works to handle the spin connection in the purely isotropic model [see the discussion above Eq. (50)]. Specifically, defining

$$\tilde{\psi}(t,x,y) = \sqrt{v_1^1(x)v_2^2(y)}\psi(t,x,y)\,, \tag{82}$$

and the variables $\xi = F_j(x), \upsilon = G_j(y)$, we can see that

$$\left[\partial_t + \hat{\sigma}^1\partial_\xi + \hat{\sigma}^2\partial_\upsilon\right]\tilde{\psi}(\xi,\upsilon) = 0\,. \tag{83}$$

We can use this to write the time-dependent wavefunction in terms of its flat space analogue. Recall [Sec. 5.2] that $\{x_i\}, \{y_j\}$ are the zeros of $v_1^1(x)$ and $v_2^2(y)$, respectively, and that $B_{ij} = (x_i, x_{i+1}) \times (y_j, y_{j+1})$. Let $\psi^0(x,y)$ be the initial wavefunction. For each $i, j$, we define

$$\psi_{flat}^{0,ij}(\xi,\upsilon) = \psi^0[F_i^{-1}(\xi), G_j^{-1}(\upsilon)] \tag{84}$$

for all $(\xi,\upsilon) \in \mathcal{R}^2$, and we let $\psi_{flat}^{ij}(t,\xi,\upsilon)$ be the flat-space time-evolution of $\psi_{flat}^{0,ij}(\xi,\upsilon)$ under the Dirac equation [Eq. (83)]. We can then write down the time-dependent wavefunction to the general xy-factored orthogonal model:

$$\psi(t,x,y) = \frac{1}{\sqrt{v_1^1(x)v_2^2(y)}}\sum_{ij}\Theta_{ij}^{xy}(x,y)\psi_{flat}^{ij}[t,F_i(x),G_j(y)]\,, \tag{85}$$

where

$$\Theta_{ij}^{xy}(x,y) \equiv \Theta(x-x_i)\Theta(x_{i+1}-x)\Theta(y-y_j)\Theta(y_{j+1}-y)\,, \tag{86}$$

and $\Theta$ is the Heaviside indicator function.

The existence of a global coordinate chart for each non-singular region allows us to write down the time dynamics in terms of the familiar flat-space Dirac massless propagation, which we can understand via classical 2D wave propagation. Each spinor component of the time-dependent wavefunction for a massless Dirac fermion on flat space satisfies the (2+1)-D classical wave equation; the corresponding Cauchy problem can be expressed simply in terms of Poisson's integral formula. Huygens' principal famously fails for even-dimensional wave equations, so the density does not strictly propagate on the light cone, but the "wake" left in the interior of the lightcone decays and the density propagates to infinity in the long run.

As a wavefront approaches a singular wall, the $F(x)$ [or $G(y)$] functions blow up, distorting its trajectory parallel to the wall and keeping it from reaching the wall, corresponding extremely well with the geodesic dynamics. As in the geodesic picture the singular walls carve the manifold up into a "multiverse" grid, in this case made up by the boxes ("universes") $B_{ij}$, and even quantum tunneling is unable to transfer quantum density from one box to another.

## 6.3 Dreibien walls

We now give a quantum treatment of the Dreibein wall models of Sec. 5.3. In this case, the spin connection vanishes and translational invariance in the $y$-direction lets us move to $y$-momentum eigenstates. The equation to solve is

$$[\hat{\sigma}_1\partial_x + ikm(x)\hat{\sigma}_2]\psi_{E,k}(x) = iE\psi_{E,k}(x), \tag{87}$$

where $k$ denotes the conserved $y$-momentum.

Squaring the Dreibein wall Dirac equation [Eq. (87)] gives the curved-space wave equations:

$$\left[\partial_x^2 - k^2m(x)^2 - km'(x) + E^2\right]\psi_{E,k}^1(x) = 0, \tag{88a}$$

$$\left[\partial_x^2 - k^2m(x)^2 + km'(x) + E^2\right]\psi_{E,k}^2(x) = 0. \tag{88b}$$

We assume $k > 0$, noting that if $k < 0$ we may simply swap $\psi^1$ and $\psi^2$.

For $E = 0$, we can always solve these with the "zero mode" solutions:

$$\psi_{0,k}^1(x) = N_{0,k}^1 \exp\left[+k\int_0^x dz\, m(z)\right], \tag{89a}$$

$$\psi_{0,k}^2(x) = N_{0,k}^2 \exp\left[-k\int_0^x dz\, m(z)\right]. \tag{89b}$$

If $m(-\infty) < 0 < m(\infty)$, the $\psi^1$ ($\psi^2$) solution is normalizable for $k < 0$ ($k > 0$). However, if $m(-\infty)$ and $m(\infty)$ have the same sign, then there are not normalizable zero modes, giving a $\mathbb{Z}_2$ topological condition on the existence of a continuum of zero-mode solutions.

Eq. (87) is a continuum-Dirac description for a 1D class BDI topological insulator, and mass twists produce zero modes as in the Su-Schrieffer-Heeger model [35]. For the two-dimensional problem studied here, the zero modes in Eqs. (89) constitute *flat-band* edge states, since a confined edge solution exists for any nonzero $k$.

### 6.3.1 Linear wall

In the linear case, $m(x) = x$, Eqs. (88) are simply harmonic oscillator Hamiltonians with frequency $k^2$ and energy $\tilde{E}_{1,2} = E^2 \pm k$, and the familiar normalizable solutions are in terms of Hermite polynomials. There is a bound state solution for $\psi^2$ if $\tilde{E}_2 = E^2 + k = 2k(n+1/2)$, which gives $E = \pm\sqrt{2nk}$. In turn, we find $\tilde{E}_1 = E^2 - k = 2k[(n-1)+1/2]$, corresponding to the preceding energy level. Building a spinor out of adjacent harmonic oscillator wavefunctions, using Eq. (87) to determine the relative coefficients, and normalizing, we find

$$\psi_{E,k}(x,y) = \frac{e^{iky}}{\sqrt{4\pi}}\left(\frac{|k|}{\pi}\right)^{1/4}\frac{e^{-|k|x^2/2}}{\sqrt{2^n n!}}\hat{\mathcal{P}}_k\begin{bmatrix}\sqrt{2n}H_{n-1}(\sqrt{|k|}x)\\ i\,\mathrm{sgn}(E)H_n(\sqrt{|k|}x)\end{bmatrix}, \tag{90}$$

where $H_n$ is the $n^{th}$ Hermite polynomial (and we set $H_{-1} = 0$ for notational convenience), $n \in \{0,1,2,3,\dots\}$, and

$$\hat{\mathcal{P}}_{s_k} = \begin{cases} \hat{1}, & k > 0, \\ \hat{\sigma}_1, & k < 0, \end{cases} \tag{91}$$

acts on the spinor space. The energy states are simply

$$E_n = \pm\sqrt{2kn}. \tag{92}$$

In particular, the whole spectrum consists of bound states. When n=0, we recover the zero-modes

$$\psi_{0,k}(x,y) = \frac{e^{iky}}{\sqrt{2\pi}} \left(\frac{|k|}{\pi}\right)^{1/4} e^{-|k|x^2/2} \hat{\mathcal{P}}_k \begin{bmatrix} 0 \\ 1 \end{bmatrix}, \tag{93}$$

which could also have been obtained from Eqs. (89).

While we do not calculate quantum dynamics in this model, the appearance of the harmonic oscillator wavefunctions in the spinor solution is highly suggestive of oscillatory motion similar to our geodesic trajectories, depicted in Fig. 12.

### 6.3.2  tanh wall

In the other case, $m(x) = \tanh(x)$, we find that the energy states take the form

$$\psi_{E,k}(x,y) = A_{E,k} e^{iky} \hat{\mathcal{P}}_k \begin{bmatrix} \sqrt{k-m}\, P_{k-1}^{-\mu}[\tanh(x)] \\ i\, \mathrm{sgn}(E)\sqrt{k+m}\, P_k^{-\mu}[\tanh(x)] \end{bmatrix}, \tag{94}$$

where $P_k^{-\mu}$ is an associated Legendre function of the first kind (technically, a "Ferrer's function"), $\mu = \sqrt{k^2 - E^2}$, $\hat{\mathcal{P}}_k$ is as in Eq. (91), and $A_{E,k}$ give normalization constants that we don't compute.

When $E^2 < k^2$, we find discrete normalizable bound states for $k - \mu \in \{0, 1, 2, \dots\}$, giving the $\lfloor k \rfloor + 1$ bound state energies

$$E = \pm\sqrt{n(2k-n)}, \qquad n \in \{0, 1, \dots, \lfloor k \rfloor\}. \tag{95}$$

At $E^2 > k^2$, there is a crossover to a continuum of scattering states described by imaginary-order Ferrer's functions. As $E \to 0$, the identity $P_k^{-k}(z) = C_k(1-z^2)^{k/2}$ (for some constant $C_k$) recovers the zero-mode solutions

$$\psi_{0,k}(x,y) = A_{0,k} e^{iky} \mathrm{sech}(x)^k \hat{\mathcal{P}}_k \begin{bmatrix} 0 \\ 1 \end{bmatrix}. \tag{96}$$

### 6.4  Circular nematic model

Finally, we consider the quantum solution of the circular nematic model of Sec. 5.4. Introducing angular-momentum eigenstates [Eq. (77)], the curved-space Dirac equation takes the form

$$\left([1+\rho(r)]\partial_r - [1-\rho(r)]\frac{l}{r} + \left[\frac{\rho'(r)}{2} + \frac{\rho(r)}{r}\right]\right)\psi_{E,l}^1(r) = iE\psi_{E,l}^2(r), \tag{97a}$$

$$\left([1+\rho(r)]\partial_r + [1-\rho(r)]\frac{l+1}{r} + \left[\frac{\rho'(r)}{2} + \frac{\rho(r)}{r}\right]\right)\psi_{E,l}^2(r) = iE\psi_{E,l}^1(r). \tag{97b}$$

### 6.4.1  Zero modes

In the zero-energy case, we can directly integrate to find

$$\psi_{E,l}^1(r) = \psi_{E,l}^1(1)\sqrt{\frac{1+\rho(1)}{1+\rho(r)}}\exp\left[\int_1^r \frac{dz}{z}\frac{l[1-\rho(z)]-\rho(z)}{1+\rho(z)}\right], \tag{98a}$$

$$\psi_{E,l}^2(r) = \psi_{E,l}^2(1)\sqrt{\frac{1+\rho(1)}{1+\rho(r)}}\exp\left[-\int_1^r \frac{dz}{z}\frac{l[1-\rho(z)]+1}{1+\rho(z)}\right]. \tag{98b}$$

Suppose for simplicity that $\rho(0) = 0$ and that asymptoically $\rho(r) \to \rho_\infty$. Near the origin, we have

$$\psi_{E,l}^1(r) \approx A r^l, \tag{99}$$

but far from the origin, where $\rho(r) \approx \rho_\infty$, we have

$$\psi_{E,l}^1(r) \approx B r^{l(1-\rho_\infty)/(1+\rho_\infty) - \rho_\infty/(1+\rho_\infty)}, \tag{100}$$

where $A, B$ are constants. We thus have distinct, normalizable zero-energy wavefunctions for $\psi_{E,l}^1(r)$ for all $l > 0$ if $\rho_\infty > 1$. That is, there are infinitely many zero-energy states if and only if there is an odd number of singularity rings, giving another $\mathbb{Z}_2$ topological condition on zero-modes. [Compare with Sec. 6.3.]

### 6.4.2  Power-law models

In the case of the power-law models, $\rho(r) = r^\alpha$, which host singular manifolds on the unit circle, we can simply evaluate Eqs. (98):

$$\psi_{E,l}^1(r) = A_1 (1 + r^\alpha)^{-(2+\alpha)/(2\alpha)} \left[ \frac{r}{(1 + r^\alpha)^{2/\alpha}} \right]^l, \tag{101a}$$

$$\psi_{E,l}^2(r) = A_2 (1 + r^\alpha)^{-(2+\alpha)/(2\alpha)} \left[ \frac{r}{(1 + r^\alpha)^{2/\alpha}} \right]^{-(l+1)}. \tag{101b}$$

The $A_i$ are normalization constants. Thus, for any $\alpha > 1$, we have a whole series of zero-energy states corresponding to different angular momenta.

To gain an understanding of the energy spectrum, we focus on the $\alpha = 2$ case, which admits a nice solution due to a symmetry under the transformation $r \leftrightarrow 1/r$. The spinor eigenfunctions are given by

$$\psi_{E,l}(\theta, r) = \frac{N_{E,l}}{r} [\mathrm{sgn}(1-r)]^n e^{il\theta} \begin{cases} \begin{bmatrix} 4(l+1)\omega^{l+1} \mathcal{F}_{2,1}\left[\frac{-n}{2}, l + \frac{1+n}{2}, 1+l, \omega^2\right] \\ ie^{i\theta} E\, \mathrm{sgn}(1-r)\omega^{l+2} \mathcal{F}_{2,1}\left[\frac{1-n}{2}, l + \frac{2+n}{2}, 2+l, \omega^2\right] \end{bmatrix}, & l \geq 0, \\ \begin{bmatrix} iE\, \mathrm{sgn}(1-r)\omega^{1-l} \mathcal{F}_{2,1}\left[\frac{1-n}{2}, \frac{n}{2} - l, 1-l, \omega^2\right] \\ e^{i\theta} 4(-l)\omega^{-l} \mathcal{F}_{2,1}\left[\frac{-n}{2}, \frac{n-1}{2} - l, -l, \omega^2\right] \end{bmatrix}, & l \leq -1, \end{cases} \tag{102}$$

where $\mathcal{F}_{2,1}(a, b; c; z)$ is the Gaussian hypergeometric function and we have introduced the variable

$$\omega(r) = \frac{2r}{1 + r^2}. \tag{103}$$

Above, the energy levels are given by

$$E = \pm 2\sqrt{n(n + |2l + 1|)}, \tag{104}$$

so that we again have a full spectrum of bound states.

## 6.5  Comparison of quantum and geodesic pictures

We have provided quantum treatments of the curved-space Dirac equation for the simple space-time geometries in Sec. (5) in order to provide intuition for the ways in which the geodesic flow captures the quantum mechanics picture. We have found several ways in which the quantum mechanics mirrors the geodesic geometry.

In the dreibein wall geometries, we saw that while all geodesics are bounded for the linear wall, only some are bounded for the tanh wall. We see this situation mirrored in the quantum spectra of these manifolds; the linear wall model hosts an entire spectrum of bound states, while the tanh wall model's spectrum features a crossover from a bound-state region to a continuum of scattering states. Similarly, we have seen that the $\alpha = 2$ version of the circular nematic power-law model has only bounded geodesics, and that its spectrum also consists entirely of discrete bound states.

On the other hand, we have seen that models that host only bounded geodesic orbits can also host continuous spectra entirely of scattering states – the isotropic power-law models and the xy-factored model both provide examples of this. However, in these cases, we are able to also look at the *quantum dynamics* of the problem, and we see that the dynamics is in extremely close correspondence with the geodesic picture.

## 7   Conclusion

The effects of "artificial" quenched gravity (as defined in the Introduction) on 2D massless Dirac carriers could have important consequences for understanding and manipulating low-dimensional Dirac materials. The action in Eq. (3b) is equivalent to a theory of massless electrons on a certain class of static, curved spacetime manifolds, described by the metric in Eq. (11c). The geometry of null geodesic trajectories is heavily affected by the presence of both isotropic and nematic curvature singularities that can arise in these spacetimes. Isotropic singularities can asymptotically capture geodesics that pass sufficiently close. On the other hand, null geodesics can traverse nematic singularity domain walls, but experience a *geodesic collimation* effect that fixes their transit velocity. These domain walls can exhibit a horizon effect, trapping null geodesics as bound states that perpetually lens back and forth across the nematic singularity line.

In a semiclassical picture of the quantum dynamics, the influence of nematic singularity walls on null geodesics presents a compelling potential mechanism for *pairing enhancement* in Dirac superconductors along these singular manifolds. On one hand, states gravitationally bound to domain wall horizons could provide a link to quasi-1D physics. The latter has been long suspected to play a role in enhancing strong correlations in quantum materials, and possibly in the mechanism for high-$T_c$ superconductivity in particular [18–21]. States gravitationally bound to the singular manifolds will feel the effects of an interaction-enhancing flat band dispersion. On the other hand, the collimation phenomenon drives particles at the same spatial location to equal or opposite momenta, a *geometric effect* reminiscent of the dynamics induced by kinematical constraints and attractive interactions in BCS theory.

If singularity walls in the spacetime manifold could enhance or even induce superconducting pairing, then quenched artificial gravity could underlie a simple, universal mechanism for gap enhancement in 2D Dirac superconductors (SCs). In the scenario where quenched gravitational disorder arises from gap fluctuations in a d-wave SC, the prevalence of nematic singularities is dictated by the *ratio* of gap fluctuations to the size of the gap. Therefore, singularities could be *more* common in a weak-pairing SC state given a fixed degree of fluctuation, possibly induced by a low-temperature pairing mechanism. Pairing enhancement due to nematic singularity walls could then create a negative feedback loop, terminating when the gap has hardened sufficiently so as to suppress singularities and their concomitant 1D bound states. This paradigm allows disorder to play a constructive role, which could offer insight into the puzzling indifference of the cuprates to dopant-induced disorder [22].

Finally, while static gravity is the focus of this paper, Eq. (36) shows that the geodesic collimation effect of nematic singularities survives a generalization to time-dependent fluctu-

ations of the Dirac cone. Since the collimation effect can provide a mechanism for interesting physics, Eq. (36) could serve as the foundation of an attempt to relate this to time-dependent fluctuations of a superconducting gap. This could connect with popular theories of strong correlation physics based on fluctuation-driven competing orders and proximate quantum critical points [36]. An approach that passes all sources of "fluctuation" (both ordered and disordered) through the intermediary step of (spatial and temporal) gap modulation has the potential to unify several competing frameworks into a single mechanism for superconductivity. Exploring these possibilities is a goal for future work.

# Acknowledgements

We thank Mustafa Amin and Ilya Gruzberg for useful conversations. This work was supported by the Welch Foundation Grant No. C-1809 and by NSF CAREER Grant No. DMR-1552327. S.M.D. further acknowledges funding from the Laboratory for Physical Sciences.

# A  Reformulation of the geodesic equation

This appendix outlines useful reformulations of the geodesic equation. First we give an angular, first-order formulation based on Eq. (29). We then explain the derivation of Eqs. (30a)–(30c).

## A.1  Angular reformulation

We can use the nullity condition, Eq. (29) with $\Delta = 0$, to reformulate the geodesic equation in terms of the spatial velocity-vector angle $\theta$:

$$\dot{x}(t) = \sigma(\theta, \mathbf{x}) \cos\theta, \tag{105a}$$

$$\dot{y}(t) = \sigma(\theta, \mathbf{x}) \sin\theta, \tag{105b}$$

$$\dot{\theta}(t) = \frac{1}{\sigma(\theta, \mathbf{x})} \left[ a_2(\mathbf{x}) \cos\theta - a_1(\mathbf{x}) \sin\theta \right], \tag{105c}$$

so that $dy/dx = \tan(\theta)$, and where the $\{a_j\}$ are taken from the right-hand side of Eq. (27), $\ddot{x}_j(t) \equiv a_j[\mathbf{x}(t)]$.

The angular formulation has a built-in error-resistance for numerical solution. By using the nullity condition to reduce the equations to first order, we guarantee that the particle moves along a null geodesic. Even as the solver inevitably accumulates errors due to the angular update, the particle may move along a different geodesic than the one it started on, but it will still be on a null geodesic.

## A.2  Tangent vector projected onto the dreibein

We can express the geodesic equation concisely in terms of the dreibein. We project the tangent (3-velocity) vector onto the dreibein,

$$u^A(s) \equiv E^A_\mu[\mathbf{x}(s)] \frac{dx^\mu}{ds} . \tag{106}$$

We can then re-express the geodesic equation as the time-evolution equation for $u$, which can be written simply in terms of $E^A_\mu$ as

$$
\begin{aligned}
\frac{du^J}{ds} &= \eta^{JM}\eta_{AP}\left[E^\mu_B E^\nu_M - E^\nu_B E^\mu_M\right](\partial_\nu E^P_\mu)u^A u^B, \\
&= \frac{1}{2}\eta^{JM}\eta_{AP}(E^\mu \wedge E^\nu)_{BM}(dE^P)_{\nu\mu}u^A u^B.
\end{aligned}
\tag{107}
$$

For null geodesics, we have

$$
\eta_{AB}u^A u^B = g_{\mu\nu}(dx^\mu/ds)(dx^\nu/ds) = 0,
\tag{108}
$$

so that we may parametrize the vector $u$ as

$$
u^A(s) \to u^0(s)\begin{bmatrix} 1 \\ \cos[\phi(s)] \\ \sin[\phi(s)] \end{bmatrix}.
\tag{109}
$$

### A.3 Derivation of Eqs. (30a)–(30c)

The special structure of the quenched gravitational metric in Eq. (11c) allows us to make further progress. In particular, using the temporal first integral equation (24) and the time-space block diagonality of the dreibein, we have

$$
\frac{dt}{ds} = E^0_0[\mathbf{x}(s)]u^0(s) = \gamma[\mathbf{x}(s)] = \left\{E^0_0[\mathbf{x}(s)]\right\}^2,
\tag{110}
$$

[see Eq. (25)], and where we have set the constant $E/m = 1$. In this equation, $E^0_0$ is the $A = 0$, $\mu = 0$ component of $E^\mu_A$, which is the inverse of the same component of $E^A_\mu$ [Eq. (106)]. We conclude that $u^0(s) = E^0_0[\mathbf{x}(s)] = 1/\sqrt{|\mathbf{v_1} \times \mathbf{v_2}|}$ [Eq. (11a)].

As before, we implement the global time reparametrization via Eq. (26). Combining this with Eq. (106) and projecting out the spatial components of the geodesic equation, we obtain

$$
\dot{\mathbf{x}}(t) = \begin{bmatrix} v^1_1(\mathbf{x}) & v^1_2(\mathbf{x}) \\ v^2_1(\mathbf{x}) & v^2_2(\mathbf{x}) \end{bmatrix}\hat{\boldsymbol{\phi}}(t) \equiv \hat{V}(\mathbf{x})\hat{\boldsymbol{\phi}}(t),
\tag{111}
$$

where $\hat{\boldsymbol{\phi}} \equiv [\cos\phi, \sin\phi]^T$ is a unit vector. This gives Eqs. (30a) and (30b), but it remains to determine the dynamics of $\phi(t)$. We may use the parametrization of $u^A$ [Eq. (109)] and the time evolution equation for $u^A$ [Eq. (107)] together to find that

$$
\frac{d\hat{\boldsymbol{\phi}}_j}{dt} = \sum_{a,b,t,l\in\{1,2\}}\left[V_{lb}V_{tj} - V_{tb}V_{lj}\right]\left(\partial_t V^{-1}_{al}\right)\hat{\boldsymbol{\phi}}_a\hat{\boldsymbol{\phi}}_b.
\tag{112}
$$

[We note that the term in brackets in Eq. (112) vanishes for most index assignments.] Backing out the implied ODE for $\phi(t)$ finally gives Eq. (30c).

## B Length-scale dependence

Let $a$ denote the length scale on which the lightcone modulations fluctuate, for example in a random quenched gravitational potential (QGD). We will extract the dependence of geodesic trajectories on $a$. Let $g^{(a)}_{\mu\nu}$ be the metric corresponding to disorder potentials fluctuating on

length scale $a$, and let $g^{(1)}_{\mu\nu}$ be the metric corresponding to the same potential, but scaled so that $a = 1$ :

$$g^{(a)}_{\mu\nu}[\mathbf{x}] = g^{(1)}_{\mu\nu}\left[\frac{\mathbf{x}}{a}\right].\tag{113}$$

Next, let $\Gamma^{(a)\rho}_{\mu\nu}$ be the Christoffel symbols corresponding to the metric $g^{(a)}_{\mu\nu}$. Since the Christoffel symbols are related to the metric via spatial derivatives, we find

$$\Gamma^{(a)\mu}_{\alpha\beta}[\mathbf{x}] = \frac{1}{a}\Gamma^{(1)\mu}_{\alpha\beta}\left[\frac{\mathbf{x}}{a}\right].\tag{114}$$

Now, let $[t_{(a)}(s), \mathbf{x}_{(a)}(s)]$ denote a solution to the geodesic equation at length scale $a$:

$$\partial^2_s x^\rho(s) + \Gamma^{(a)\rho}_{\mu\nu}[\mathbf{x}][\partial_s x^\mu(s)][\partial_s x^\nu(s)] = 0.\tag{115}$$

The variable transformations $s = a\tilde{s}$, $x^\mu = a\tilde{x}^\mu$ and Eq. (114) map Eq. (115) to

$$\frac{1}{a}\partial^2_{\tilde{s}}\tilde{x}^\rho(\tilde{s}) + \frac{1}{a}\Gamma^{(1)\rho}_{\mu\nu}[\tilde{\mathbf{x}}][\partial_{\tilde{s}}\tilde{x}^\mu(\tilde{s})][\partial_{\tilde{s}}\tilde{x}^\nu(\tilde{s})] = 0.\tag{116}$$

Thus, if $[t_{(a)}, \mathbf{x}_{(a)}]$ is a geodesic with metric length scale $a$, then $[\tilde{t}, \tilde{\mathbf{x}}] = (1/a)[t_{(a)}, \mathbf{x}_{(a)}] = [t_{(1)}, \mathbf{x}_{(1)}]$ is a solution with length scale $a = 1$, so that geodesics on different length scales are related by a simple inflation transformation.

## C  Other submodels: diagonal and off-diagonal

The pure isotropic and nematic models introduced in Sec. 4 are studied alongside the general QGD Hamiltonian in the quantum setting via numerical exact diagonalization in Ref. [17]. In that paper, these are referred to as models "c" and "d," respectively. That work also introduces two other models ["a," "b"]. While these models aren't of primary interest for us here in light of the pseudospin invariance of Sec. 2.4, we introduce them here and present some properties of their geodesics, for comparison with Ref. [17].

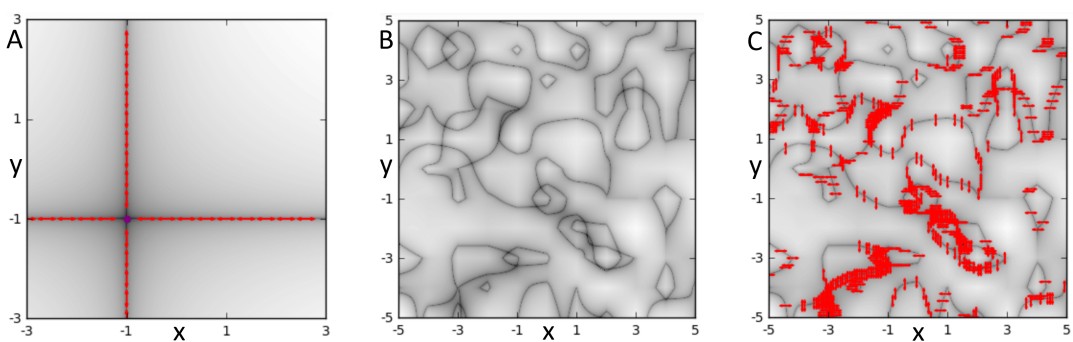

Figure 15: Curvature in the pure diagonal model. A: Heat map of the time-dilation factor $\gamma$ [Eq. (118)] for the "disorder space" diagonal manifold, with $\delta v^1_1 = x$ and $\delta v^2_2 = y$. We note the singularities lie along the lines $\{x = -1, \ y = -1\}$. B: A heat map of the time-dilation factor for a random realization of purely diagonal QGD. C: The heat map in B annotated to mark the collimation angles of the nematic singularities, which we note are all either horizontal or vertical.

## C.1 Pure diagonal model

The pure diagonal model is defined by the disorder vectors

$$\mathbf{v_1}(\mathbf{x}) = \begin{bmatrix} 1 + \delta v_1^1(\mathbf{x}) \\ 0 \end{bmatrix}, \tag{117a}$$

$$\mathbf{v_2}(\mathbf{x}) = \begin{bmatrix} 0 \\ 1 + \delta v_2^2(\mathbf{x}) \end{bmatrix}. \tag{117b}$$

This corresponds to "model a" in Ref. [17]. We note that by pseudospin invariance, its properties generalize to all models with $\mathbf{v_1} \cdot \mathbf{v_2} = 0$ everywhere. The time-dilation factor is given by

$$\gamma(\mathbf{x}) = \frac{1}{|(1 + \delta v_1^1(\mathbf{x}))(1 + \delta v_2^2(\mathbf{x}))|}, \tag{118}$$

so that we have a nematic singularity when either $\delta v_1^1 = -1$ or $\delta v_2^2 = -1$, and an isotropic singularity when both $\delta v_1^1 = \delta v_2^2 = -1$. In this model, all isotropic singularities lie at an intersection of nematic singularity manifolds—see Fig. 15. The squared speed-of-light in Eq. (29) (with $\Delta = 0$) reduces to

$$\sigma^2(\theta, \mathbf{x}) = \frac{|(1 + \delta v_1^1(\mathbf{x}))(1 + \delta v_2^2(\mathbf{x}))|^2}{(1 + \delta v_1^1(\mathbf{x}))^2 \sin^2 \theta + (1 + \delta v_2^2(\mathbf{x}))^2 \cos^2 \theta}. \tag{119}$$

We see from Eq. (119) that the geodesic collimation effect takes a simple form for these models: if the singularity corresponds to $\delta v_1^1 = -1$ ($\delta v_2^2 = -1$), then the geodesic may only pass through vertically (horizontally).

## C.2 Pure off-diagonal model

The pure off-diagonal model is defined by

$$\mathbf{v_1}(\mathbf{x}) = \begin{bmatrix} 1 \\ v_2^1(\mathbf{x}) \end{bmatrix}, \tag{120a}$$

$$\mathbf{v_2}(\mathbf{x}) = \begin{bmatrix} v_1^2(\mathbf{x}) \\ 1 \end{bmatrix}. \tag{120b}$$

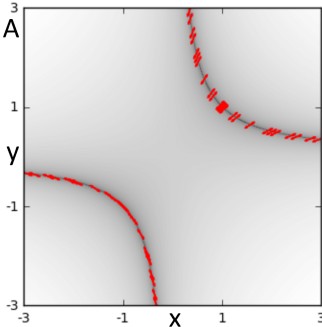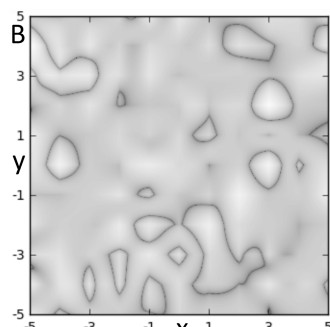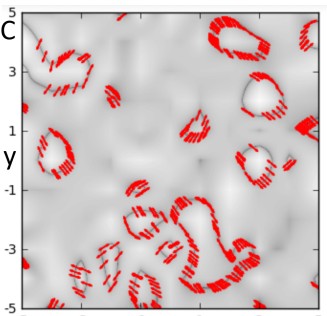

Figure 16: Curvature in the pure off-diagonal model. A: Heat map of time-dilation factor $\gamma$ [Eq. (121)] for the "disorder space" off-diagonal manifold, with $v_{12} = x$ and $v_{21} = y$. We note the singularities lie along the hyperbola $\{xy = 1\}$. B: A heat map of time-dilation factor for a random realization of purely off-diagonal QGD. C: The heat map in B annotated to mark the collimation angles of the nematic singularities, which we note are clustered around $\pm \pi/4$.

This corresponds to "model b" in Ref. [17], and we note that it is equivalent to the pure nematic model by pseudospin invariance. The time-dilation factor is given by

$$\gamma(\mathbf{x}) = \frac{1}{|1 - v_1^2(\mathbf{x})v_2^1(\mathbf{x})|}, \tag{121}$$

so that we have singularities whenever $v_1^2 v_2^1 = 1$. The constant-interval speed condition [Eq. (29)] reduces to (for null geodesics with $\Delta = 0$)

$$\sigma^2(\theta, \mathbf{x}) = \frac{|v_{12}(\mathbf{x})v_{21}(\mathbf{x}) - 1|^2}{[\sin\theta - v_{21}(\mathbf{x})\cos\theta]^2 + [\cos\theta - v_{12}(\mathbf{x})\sin\theta]^2}. \tag{122}$$

We can parametrize a point on the singularity manifold by $(v_1^2, v_2^1) = (\cot\phi, \tan\phi)$ [Fig. (16)]. Eq. (122) then shows that a geodesic can pass through a curvature singularity if and only if it is at the angles $\phi$ (or $\phi + \pi$), relating the collimation angles to the geometry of the singularity manifold in $\{v_1^2, v_2^1\}$-space.

# D    Additional details on quantum treatments of toy models

This appendix collects additional details of the quantum solutions presented in Sec. 6.

## D.1    Isotropic power law model

Using the angular momentum states [Eq. (77)] and specializing to the power-law case, $v(r) = r^\alpha$, Eq. (76) becomes

$$r^\alpha \left[ \partial_r + \frac{l+1}{r} \right] \tilde{\psi}_{l,E}^2(r, \theta) = iE\tilde{\psi}_{l,E}^1(r, \theta), \tag{123a}$$

$$r^\alpha \left[ \partial_r - \frac{l}{r} \right] \tilde{\psi}_{l,E}^1(r, \theta) = iE\tilde{\psi}_{l,E}^2(r, \theta). \tag{123b}$$

Iterating these Dirac equation, we find that the spinor components satisfy the associated second-order curved-space wave equations

$$r^{2\alpha} \left[ \partial_r^2 + \left( \frac{1+\alpha}{r} \right) \partial_r - \frac{l(l+\alpha)}{r^2} \right] \tilde{\psi}_{l,E}^1(r) = -E^2 \tilde{\psi}_{l,E}^1(r), \tag{124a}$$

$$r^{2\alpha} \left[ \partial_r^2 + \left( \frac{1+\alpha}{r} \right) \partial_r - \frac{(l+1-\alpha)(l+1)}{r^2} \right] \tilde{\psi}_{l,E}^2(r) = -E^2 \tilde{\psi}_{l,E}^2(r). \tag{124b}$$

For $\alpha \neq 1$ he general solutions are

$$\tilde{\psi}_{l,E}^1(r) = \frac{1}{r^{\alpha/2}} \left( A^1 \mathcal{J}_m \left[ \frac{Er^{1-\alpha}}{1-\alpha} \right] + B^1 \mathcal{J}_{-m} \left[ \frac{Er^{1-\alpha}}{1-\alpha} \right] \right), \tag{125a}$$

$$\tilde{\psi}_{l,E}^2(r) = \frac{1}{r^{\alpha/2}} \left( A^2 \mathcal{J}_{m-1} \left[ \frac{Er^{1-\alpha}}{1-\alpha} \right] + B^2 \mathcal{J}_{1-m} \left[ \frac{Er^{1-\alpha}}{1-\alpha} \right] \right). \tag{125b}$$

Where $m$ is given by Eq. (79) and $\mathcal{J}_m$ is a Bessel function of order $m$. Only the solution with $m > 1/2$, (which is equivalent to $l > -1/2$) is normalizable and physical. (In the cases $\alpha \in \{0, 2\}$, $m$ is integral, and the non-physical solution is a Neumann function.)

Plugging the normalizable solutions of Eqs. (125) back into Eqs. (76) then determines the relative spinor structure. Restoring the $1/r^{\alpha/2}$ prefactor from spin connection removal [see Eq. (49)] and providing normalization finally gives Eq. (78).

To see that states with equal angular momenta and distinct energy are orthogonal, note that [with $z = r^{1-\alpha}/(1-\alpha)$]

$$\left(\frac{1}{1-\alpha}\right)\int_0^\infty dr\, r\frac{1}{r^{2\alpha}}\mathcal{J}_m\left[\frac{Er^{1-\alpha}}{1-\alpha}\right]\mathcal{J}_m\left[\frac{E'r^{1-\alpha}}{1-\alpha}\right] = \int_0^\infty dz\, z\mathcal{J}_m[Ez]\mathcal{J}_m[E'z] \qquad (126a)$$

$$= \frac{1}{E}\delta[E-E']. \qquad (126b)$$

### D.2 tanh wall

In the case $m(x) = \tanh(x)$, Eqs. (88) take the form

$$\partial_x^2\psi_{E,k}(x) = \left[k^2\tanh(x)^2 + \hat{\sigma}^3 k\,\text{sech}(x)^2 - E^2\right]\psi_{E,k}(x). \qquad (127)$$

The substitution $z = \tanh(x)$ actually turns these exactly into generalized Legendre equations:

$$\left[\partial_z(1-z^2)\partial_z + k(k-\hat{\sigma}_3) - \frac{k^2-E^2}{1-z^2}\right]\psi_{E,k}(z) = 0, \qquad (128)$$

and the solutions can generally be expressed in terms of the associated Legendre functions (specifically "Ferrer's functions of the first kind"):

$$\psi_{E,k}^1 = A_1 P_{k-1}^{-\mu}[\tanh(x)] + B_1 P_{k-1}^{\mu}[\tanh(x)], \qquad (129a)$$

$$\psi_{E,k}^2 = A_2 P_k^{-\mu}[\tanh(x)] + B_2 P_k^{\mu}[\tanh(x)], \qquad (129b)$$

where $\mu = \sqrt{k^2-E^2}$. (If $\mu$ is an integer, we must replace $P_k^\mu$ with $Q_k^\mu$, the Ferrer's function of the second kind, but we'll see that this is unimportant.)

For $k^2 > E^2$, $\mu \geq 0$, and only $P_k^{-\mu}[\tanh(x)]$ is normalizable at $x \to \infty$. At $x \to -\infty$, we use the identity

$$P_k^{-\mu}(-z) = \cos\left[\pi(k-\mu)\right]P_k^{-\mu}(z) - \frac{2}{\pi}\sin\left[\pi(k-\mu)\right]Q_k^{-\mu}(z) \qquad (130)$$

and the fact that $Q_k^{-\mu}(z)$ diverges at $z = 1$ for $\mu \in [0,k]$ to find the bound state energy condition $n \equiv k - \mu \in \{0,1,2,\dots\}$, giving the spectrum in Eq. (95). Enforcing compatibility of (the normalizable part of) Eqs. (129) with the original curved-space Dirac equation [Eq. (87) gives the relative spinor prefactors, yielding Eq. (94) in the main text.

### D.3 Circular nematic model: $\alpha = 2$

The curved-space Dirac equation follows from Eq. (97) by setting $\rho(r) = r^2$. In this case, the spin connection can be removed [see Eq. (50)] by setting

$$\psi_{E,l}(r) = \frac{1}{r}\frac{2r}{1+r^2}\tilde{\psi}_{E,l}(r). \qquad (131)$$

We find the simplified Dirac equation

$$\left[(1+r^2)\partial_r - (1-r^2)\frac{l}{r}\right]\tilde{\psi}_{E,l}^1(r) = \tilde{\psi}_{E,l}^2(r), \qquad (132a)$$

$$\left[(1+r^2)\partial_r + (1-r^2)\frac{l+1}{r}\right]\tilde{\psi}_{E,l}^2(r) = \tilde{\psi}_{E,l}^1(r). \qquad (132b)$$

Iterating and manipulating this, we find the curved-space wave equation for $\psi^1$:

$$\left(r + \frac{1}{r}\right)\left\{(r\partial_r)^2 - l\left[(l+1)\left(\frac{r-1/r}{r+1/r}\right) - 1\right]\right\}\tilde{\psi}^1_{E,l}(r) = -E^2\tilde{\psi}^1_{E,l}(r). \tag{133}$$

The related equation for the $\tilde{\psi}^2$ component is obtained from Eq. (133) by sending $l \leftrightarrow -(l+1)$. Eq. (133) possesses a remarkable symmetry under $r \leftrightarrow 1/r$, and this aids in the determination of the physical energy levels. To take advantage of this, we introduce the auxiliary inversion-invariant variable

$$\omega = \frac{2r}{1+r^2}, \tag{134}$$

which gives bijective maps from both $r \in [0,1]$ and $r \in [1,\infty]$ to $\omega \in [0,1]$. Re-writing Eq. (133) in terms of $\omega$ and solving, we find the 2D solution space is spanned by the functions

$$K_+(\omega) = \omega^l \mathcal{F}_{2,1}\left[\frac{1+l-\beta}{2}, \frac{\beta+l}{2}; 1+l; \omega^2\right], \tag{135a}$$

$$K_-(\omega) = \omega^{-l} \mathcal{F}_{2,1}\left[\frac{1-l-\beta}{2}, \frac{\beta-l}{2}; 1-l; \omega^2\right], \tag{135b}$$

where only the first (second) solution is normalizable near $\omega \approx 0$ for $l \geq 0$ ($l \leq -1$).

For now set $l \geq 0$. While $\psi^1(r)$ is a multiple of $K_+(\omega)$ on the intervals $[0,1]$ and $[1,\infty]$, some care must be taken in gluing these two solutions together. Writing

$$\tilde{\psi}^1_{E,l} = \begin{cases} AK_+(\omega), & r < 1, \\ BK_+(\omega), & r > 1, \end{cases} \tag{136}$$

then continuity of $\tilde{\psi}^1_{E,l}(r)$ and $\partial_r\tilde{\psi}^1_{E,l}(r)$ at $r = 1$ gives the two conditions

$$(A-B)\mathcal{F}_{2,1}\left[\frac{1+l-\beta}{2}, \frac{\beta+l}{2}; 1+l; 1\right] = 0, \tag{137a}$$

$$(A+B)(l+1-\beta)\mathcal{F}_{2,1}\left[\frac{1+l+\beta}{2}+1, \frac{\beta+l}{2}+1; 2+l; 1\right] = 0. \tag{137b}$$

So we see that $A = \pm B$, and energy states naturally fall into $\pm 1$ eigenstates under the inversion operation. Since $\beta \in (l+1, \infty)$, we can set $\beta = l+1+n$. Eqs. (137) then have an even-inversion solution ($A = B$) if $n \in \{0, 2, 4, \dots\}$ and have an odd-inversion solution ($A+B = 0$) if $n \in \{1, 3, 5, \dots\}$.

Repeating the above analysis for $l \leq -1$, and then again for the $\tilde{\psi}^1$ solution, we find the energy spectrum in Eq. (104) in the main text. Further, the solutions for $\tilde{\psi}^1, \tilde{\psi}^2$ are always compatible with the same energy spectrum, though they have opposite parity under the inversion transformation.

Enforcing compatibility with Eqs. (97) then gives the relative prefactors of the spinor components. Finally, restoring the prefactor [Eq. (131)] gives Eq. (102) in the main text.

# E Quantum dynamics for $\alpha = 2$ isotropic power-law model

We will asymptotically solve the Cauchy problem (for $\alpha = 2$), showing that all probability density of an arbitrary "away from zero" initial wavefunction eventually flows into the singularity. We let $\psi_0(r, \theta)$ be an initial wavefunction at $t = 0$. We will assume that the quantum density starts out away from zero – specifically we assume that there is an $\eta > 0$ so that $\psi_0(r, \theta) = 0$

for $r < \eta$. Finally we let $\epsilon > 0$ and $\delta > 0$ be given. We will show that there exists a time $t^*$ so that for all $t > t^*$

$$\int_0^{2\pi} d\theta \int_\epsilon^\infty dr\, r\, |\psi(t, r, \theta)|^2 < \delta\,. \tag{138}$$

We may formally solve the Cauchy problem via an eigenfunction [Eq. (78)] expansion:

$$\psi^\alpha(t, r, \theta) = \int_0^{2\pi} d\phi \int_0^\infty d\rho\, \rho\, \left( \sum_l \int_\infty^{-\infty} dE\, E\, e^{-iEt} \psi_{l,E}^{(\alpha)}(r, \theta) \psi_{l,E}^{(\alpha)}(\rho, \phi)^\dagger \right) \psi_0(\rho, \phi)$$

$$\equiv \int_0^{2\pi} d\phi \int_0^\infty d\rho\, \rho\, \mathcal{G}^\alpha(t; r, \theta; \rho, \phi) \psi_0(\rho, \phi), \tag{139}$$

where the term in large brackets defines the retarded Green's function. (Convolution with the retarded Green's function gives time evolution of an initial wavefunction.)

We note that fixing $\alpha = 2$ in Eq. (78) gives energy and angular momentum eigenstates that can be related to the eigenstates of the flat-space [$\alpha = 0$] Dirac equation:

$$\psi_{l,E}^{\alpha=2}(r, \theta) = \left[ \frac{|E|}{4\pi} \right]^{1/2} \frac{e^{il\theta}}{r^2} \left[ \begin{array}{c} \mathcal{J}_{l+1}[E/r] \\ ie^{i\theta} \mathcal{J}_l[E/r] \end{array} \right] \tag{140a}$$

$$= \frac{M(\theta)}{r^2} \left[ \frac{|E|}{4\pi} \right]^{1/2} e^{il\theta} \left[ \begin{array}{c} \mathcal{J}_l[E/r] \\ ie^{i\theta} \mathcal{J}_{l+1}[E/r] \end{array} \right] = \frac{M(\theta)}{r^2} \psi_{l,E}^{\alpha=0}\left( \theta, \frac{1}{r} \right), \tag{140b}$$

where

$$M(\theta) = \left[ \begin{array}{cc} 0 & -ie^{-i\theta} \\ ie^{i\theta} & 0 \end{array} \right]. \tag{141}$$

It follows the $\alpha = 2$ and the $\alpha = 0$ retarded Green's functions are related:

$$\mathcal{G}^{(\alpha=2)}(t; r, \theta; \rho, \phi) = \frac{1}{r^2 \rho^2} M(\theta) \mathcal{G}^{(\alpha=0)}\left( t; \frac{1}{r}, \theta, \frac{1}{\rho}; \phi \right) M(\phi). \tag{142}$$

We can use this trick to simply compute the time-dependent wavefunction for the $\alpha = 2$ time evolution in terms of its flat space counterpart. Using Eq. (142) and the change of variables $z = 1/\rho$ in Eq. (139), we have

$$\psi^{(\alpha=2)}(t, r, \theta) = \int_0^{2\pi} d\phi \int_0^\infty d\rho\, \rho\, \mathcal{G}^{(\alpha=2)}(t; r, \theta; \rho, \phi) \psi_0(\rho, \phi), \tag{143a}$$

$$= \frac{M(\theta)}{r^2} \int_0^{2\pi} d\phi \int_0^\infty dz\, z\, \mathcal{G}^{(\alpha=0)}\left( t; \frac{1}{r}, \theta, z, \phi \right) \Upsilon_0(z, \phi) \tag{143b}$$

$$= \frac{M(\theta)}{r^2} \Upsilon\left( t, \frac{1}{r}, \theta \right), \tag{143c}$$

where $\mathcal{G}^{\alpha=0}$ is the Green's function for flat space massless Dirac propagation,

$$\Upsilon_0(z, \phi) = \frac{M(\phi)}{z^2} \psi_0\left( \frac{1}{z}, \phi \right) \tag{144}$$

is a new effective "initial" wavefunction, and $\Upsilon$ is its time evolution in 2D flat space. As long as $\psi_0$ decays sufficiently quickly at infinity, $\Upsilon_0$ will remain finite as $z \to 0$. Further, $\Upsilon_0$ inherits compact support from the "away from zero" assumption; since $\psi_0 = 0$ for $\rho < \eta$, $\Upsilon_0 = 0$ for $z > 1/\eta$.

In this effective flat-space picture, all our initial data resides in a region within $1/\eta$ from the origin and in the long time limit it will propagate out towards infinity. Thus, by classical wave equation theory, there is a time $t^*$ such that the density ($|\Upsilon|^2$) left within a distance $1/\epsilon$ from the origin is less than $\delta$. Finally, for $t > t^*$, we then have

$$\int_0^{2\pi} d\theta \int_\epsilon^\infty dr\, r\, |\psi(t,r,\theta)|^2 = \int_0^{2\pi} d\theta \int_0^{1/\epsilon} dz\, z\, |\Upsilon(t,z,\theta)|^2 < \delta, \qquad (145)$$

as desired.

We note that this argument does not quite go through for the general $\alpha$ case because the orders of the Bessel functions are not integral for $\alpha \neq 2$ and so the Green's function cannot be identified with its flat space counterpart. However, since the single-particle eigenstates are qualitatively similar for all $\alpha > 1$, it seems likely that the quantum dynamics pictures are similar.

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
