# Peer review of "Geodesic geometry of 2+1-D Dirac materials subject to artificial, quenched gravitational singularities"

_SciPost Physics, doi:SciPost Phys. 12, 204 (2022)_

## Round 1 · Referee Report · Seyed Akbar Jafari (Referee 1) · 2021-10-9

Strengths

1- Introduction of the paper is nicely written and the basic message of the paper is clearly communicated. 2- Content of equations are illustrated with informative figures 3- The language of geometry ("gravitation") adopted in this paper is a new lens through which intuitive picture of classical dynamics can be constructed.

Weaknesses

1- The eventual goal of any mathematical modeling is to compute a measurable quantity. The (null) geodesics studied in this paper are not directly measurable. A simple estimate of conductivity and the role of attractors and collimator singularities in conductivity is missing.

2- As pointed out in the introduction, the null geodesics play a role in shaping the quantum mechanical wave function. Authors have meticulously computed null geodesics for interesting solvable toy models. But there is no comment/thoughts about possible roles of these (classical) solutions for the corresponding quantum mechanical problem.

Report

Summary of the paper:

The paper looks at the disorder in high temperature superconducting compounds from a theoretically appealing angle and cosiders it as QGD that arises from the modulation of the velocity of Dirac electrons. Authors view the band flattening in twisted bilayer graphene as isotropic curvature singularity. Apart from the possible role that the null geodesics play in shaping the quantum mechanical wave function in the curved spacetime, the paper provides adequately appealing semiclassical picture of the physics of disorder in the above compounds. Models for anisotropic velocity fluctuations as well as nematic fluctuations are presented and solved for their geodesics. The authors demonstrate that nematic curvature singularities collimate the geodesics, while the in the case of isotropic fluctuations, the geodesics are captures by the singularities. The fact that such singularities are like absorbers of geodesics is nicely demonstrated by a more generic solvable models.

The message of the paper is clearly communicated with a nice introduction and informative pictures. Geometric approaches are powerful and can shed new light on the physics of disorder in Dirac electrons. So I like the paper quite a lot and I am happy to recommend it for publication in SciPost. However, I do have the following comments/suggestions, the incorporation of which I hope will instructive for the paper.

Requested changes

Warnings issued while processing user-supplied markup:

  • Inconsistency: Markdown and reStructuredText syntaxes are mixed. Markdown will be used.
    Add "#coerce:reST" or "#coerce:plain" as the first line of your text to force reStructuredText or no markup.
    You may also contact the helpdesk if the formatting is incorrect and you are unable to edit your text.

Scientific comments/suggestions:

  • In Eq. (27), the conservation of E/m is associated with the global Killing vector (1,0,0)^T. Can it be explicitly derived by writing down the Killing equation? What I am wondering at this stage is that, are there any other Killing vectors in addition to the one that gives the conservation of E/m (energy)?

  • Rewriting the equation of motion in terms of t, rather than the affine parameter s in Eq. (3), seems to obscure the fact that energy is conserved as there appear quadratic dissipation-looking terms. Can the authors help the reader and the present referee to understand how the dissipation-looking terms at the end do no harm to the conservation of E/m by adding a brief calculation of intuitive explanation?

  • What is missing in this paper, is the implications of collimination and absorption of geodesics in transport properteis? What can be implied e.g. about the conductivity tensor for the nematic and isotropic singularities?

  • Classically geodesics of isotropic singularities are absorbed by the singularities. Would that quantum mechanically correspond to possible bound state? If yes, does it imply that isotropic singularities are more efficient in forming Anderson insulator?

  • How essential is the temporal-flatness condition in collimation property of nematic singularities and/or attractive property of isotropic singularities?

  • The motivation for the geometrical models of velocity modulated Dirac equation comes from hight Tc superconducting compunds. Also there is a brief mention of graphene and twisted bilayer graphene. In these examples the gravitational (geometrical) coupling can be encoded into spatial-spatial compoents of the stress tensor.

It has been recently proposed in [arxiv:2108.08183] that in 8Pmmn borophene, substitution of boron atoms with carbon atoms substantially affects the tilt of the resulting Dirac cone. Since the tilting of the Dirac cone always induces a velocity anisotropy (like nemacity), how likely are these compounds to realize possible QGD in non-superconducting phase? This will correspond to modulation of the spatio-temporal components of the velocity. Does the random modulation of tilt velocity relax "temporal flatness" condition?

Styling comments:

  • All over sectoin 1.1 the concept of "Null geodesics" has been used repeatedly, which is a crucial concept to understand the paper. Perhaps it will help the readers to define it right at the beginning.

  • Fig. 9B seems too crowded to follow examples of geodesics that collide or do not collide with the singularities. Maybe it helps the readers to make one curve of each category bolder than the others to assist the readers to follow at lest two bold geodesics.

---

## Round 1 · Referee Report · Anonymous (Referee 2) · 2021-11-1

Report

During the last years, analogies between the behavior of electrons in quantum materials and those in curved spacetimes have been extensively investigated. The presented manuscript is devoted to the problem along these lines. The authors rely on a mathematical correspondence between electrons in Dirac materials and the problem of particles’ propagation in the gravity context. The manifolds under consideration possess singularities classified into two types. The so-called nematic singularities are of particular interest due to the geodesic collimation effect reminiscent of some features known in BCS theory.

Although the primary motivation for the paper was dictated by the potential applicability of the results in the context of quantum materials, it could be of interest in the gravity context as well. For instance, the authors could have analyzed whether some known metrics (e.g., for black holes, cosmological horizons, etc.) belong to the class considered in the paper, providing thus new links between gravity and condensed matter theory. Perhaps, it will be done and published elsewhere.

---

## Round 2 · Author Response

We thank the referees for their thoughtful analysis of our work and their proposals for its improvement.
We divide the referees’ concerns and suggestions into two categories: “major” will denote concerns requiring additional nontrivial content in the paper, while “minor” will refer to content questions or small changes in the manuscript. The concerns we regard as major are the following
1. S.A.J.’s request for an estimate of conductivity,
2. S.A.J.’s request for more information on the relation of the fully-quantum picture to the geodesic flow
3. The request by referee 2 to discuss the applicability of our results and methods to important spacetime metrics in cosmology.
We first respond to these major concerns:
1. Quoting S.A.J.:

"The eventual goal of any mathematical modeling is to compute a measurable quantity. The (null) geodesics studied in this paper are not directly measurable. A simple estimate of conductivity and the role of attractors and collimator singularities in conductivity is missing."

"What is missing in this paper, is the implications of collimination and absorption of geodesics in transport properteis? What can be implied e.g. about the conductivity tensor for the nematic and isotropic singularities?"

We agree that the eventual computation of a measurable quantity (e.g., conductivity) is the ultimate goal and a numerical treatment of diffusion exponents and effects of QGD on conductivity remain a future research goal. Indeed, early numerical results show that weak QGD give diffusive behavior of geodesics, with average distance from the origin scaling as the square root of time, as expected. We hope to determine the effects of singularity proliferation on this as QGD disorder strength is enhanced to O(1); however, accurately extracting the effects of the singularities has turned out to be a surprisingly non-trivial technical issue due to a certain unexpected phenomena where geodesics can become trapped in extremely high-frequency meta-stable orbits. Treating these correctly seems to require an extremely small time-step. Over the time scales needed to extract the asymptotic diffusive character of QGD, these trajectories cannot be ignored. Further, throwing these trajectories out would ignore important data on the true effects of singularities on diffusion of geodesics. While we hope to overcome this issue in the future, the difficulties we encountered here have led us to publish this first work focusing on the interesting elementary geometric effects of the singularities and geodesics, without a heavy discussion of numerical simulation results.

We note that effects of weak geodesic scattering in kinetic theory have been considered before in other context [1], but since our focus is on the effects of the singularities formed at strong disorder, we feel a treatment like this would be too far from our central message to include in the current manuscript.

For these reasons, we would like to avoid getting into a discussion of the influence of QGD singularities on conductance, and leave this topic for future work.

2. Quoting S.A.J.:
"As pointed out in the introduction, the null geodesics play a role in shaping the quantum mechanical wave function. Authors have meticulously computed null geodesics for interesting solvable toy models. But there is no comment/thoughts about possible roles of these (classical) solutions for the corresponding quantum mechanical problem."
"Classically geodesics of isotropic singularities are absorbed by the singularities. Would that quantum mechanically correspond to possible bound state? If yes, does it imply that isotropic singularities are more efficient in forming Anderson insulator?"

We strongly agree that the discussion of geodesic dynamics as a proxy for semiclassics raises the question of how well the null geodesics capture the fully-quantum mechanical behavior. While we explain in the introduction that null geodesics give the bicharacteristics of the curved-space Dirac equation and that this is canonically associated with a ‘geometric optics’ limit in the general wave equation literature, we agree that it would be nice to analyze the fully-quantum picture on the toy models for which we are able to solve the geodesic dynamics.

To this end, we add a new section to our paper. Sec. 6 (6 pages) now presents fully-quantum calculations for all of the toy models considered in Sec. 5, and qualitatively compares the solutions with the associated geodesic flows. In two cases where we can make analytical statements about the asymptotic quantum dynamics, we find near full agreement with the geodesic solution. In the cases where quantum dynamics aren’t tractable, we instead calculate the wave functions and energy spectra explicitly. We find that in some cases, the quantum spectrum mirrors the geodesic behavior. For example: the linear dreibein wall model has only bound-state geodesics, and we find that its spectrum consists of discrete bound-state wavefunctions. On the other hand, the tanh wall model hosts both bound and unbound geodesics, and its quantum spectrum contains a transition from discrete bound states to a continuum of scattering states. Sec. 6 gives a more in-depth commentary on these features.

To specifically answer the question about isotropic singularities: we find that in the isotropic power-law models we study, the spectrum consists of a continuum of scattering states, as opposed to discrete bound states like we find in other models. While it is perhaps surprising that (for \alpha > 1) the geodesics all form bound orbits but the spectrum consists only of scattering states, we are able to show for the \alpha = 2 example that for an arbitrary initial wavefunction, the probability density asymptotically collects at the singularity, mirroring the geodesics. Since the relationship between bound states, quantum dynamics, and geodesic flow seems to be quite complex, we don’t make any statement about possible connections to Anderson localization.

Sec. 6 is supported by two additional appendices providing calculational details; Appendix D (3 pages) merely provides details to understand the derivations of energy eigenstates in Sec. 6. Appendix E (2 pages) provides the proof of our statement that in the \alpha = 2 isotropic power-law model, quantum density asymptotically approaches the singularity at the origin.

We stress that while this adds some substantial length to the paper, and while some of these results are quite interesting, these new results all play the supporting role of allowing comparison of our primary results (geodesics) with some fully-quantum results, as requested. We do not think this foray into QM changes the scope, mission, context, or conclusions of our paper.

3. Quoting Referee 2:
"Although the primary motivation for the paper was dictated by the potential applicability of the results in the context of quantum materials, it could be of interest in the gravity context as well. For instance, the authors could have analyzed whether some known metrics (e.g., for black holes, cosmological horizons, etc.) belong to the class considered in the paper, providing thus new links between gravity and condensed matter theory. Perhaps, it will be done and published elsewhere."

We agree that applications of gravity analogies to condensed matter systems could possibly reversed in a way that hopefully lets condensed matter shed light on topics in cosmology.
In this case, essentially all of the geometric features we find follow from temporal flatness [Eq.(8)], including both the nature of the singularities and their collimating effects on geodesics. The temporal flatness condition, which fixes the time-time dreibein in terms of the metric determinate, arises as a necessary condition for metrics whose Hamiltonians can be realized as Dirac-type Hamiltonians in flat space. We do not expect important cosmological metrics to have this feature: for example, the Schwarzchild metric is not temporally flat. Further, this puts some naïve limits on the ability of Hamiltonian systems in flat space to simulate many cosmologically important spacetime metrics
In addition to temporal flatness, our results also rely heavily on time-independence of the metric, (2+1)-dimensionality, and time-space block-diagonality, which all further remove our current results from cosmological relevance.
We’ve added a comment on these limitations and connections to our introduction.

Now we address minor concerns:
4. Quoting S.A.J.:
" In Eq. (27), the conservation of E/m is associated with the global Killing vector (1,0,0)^T. Can it be explicitly derived by writing down the Killing equation? What I am wondering at this stage is that, are there any other Killing vectors in addition to the one that gives the conservation of E/m (energy)?"

The (1,0,0)^T global Killing vector follows from the metric’s independence on time – it is a general fact that if the spacetime metric is independent of a coordinate xk, then the unit-vector eµ = δµk is a global Killing vector associated with the manifold’s translation symmetry in the xk coordinate. So the existence of the global Killing vector isn’t a nontrivial result.
Our comment about the Killing vector is a technical aside that isn’t really used in the paper, so we have restructured these remarks to clarify.
Since we don’t generally make assumptions about the dependence of the disorder vectors in the xy-plane, there are not additional global Killing vectors that apply to the general model. However, global killing vectors may be found in our toy models where translational or rotational invariance is present, and there will correspond to conservation of momentum or angular momentum, respectively. However, since our treatment there is based on a direct solution of the geodesic equation, we don’t bother introducing the formal Killing vector analysis.

5. Quoting S.A.J:
"Rewriting the equation of motion in terms of t, rather than the affine parameter s in Eq. (3), seems to obscure the fact that energy is conserved as there appear quadratic dissipation-looking terms. Can the authors help the reader and the present referee to understand how the dissipation-looking terms at the end do no harm to the conservation of E/m by adding a brief calculation of intuitive explanation?"

Indeed, the re-parametrization of the geodesic equation by the global time coordinate introduces friction/dissipation terms. We emphasize that this is a general feature of global-time re-parametrization – see the discussion in Wikipedia (section 3): https://en.wikipedia.org/wiki/Geodesics_in_general_relativity .
As discussed above in point (4), the geodesics have energy as a “constant of motion”. However, what this means is essentially limited to Eq.(26). When we re-parametrize in terms of global coordinate time, E plays no role in determining the trajectory of the null geodesic. [This is not true for massive geodesics – see Eq.(32)]. This is analogous to the familiar situation regarding the trajectories of light beams in GR; no knowledge of their wavelength enters a classical calculation.
More importantly, the friction/dissipation terms play a key role in the geodesics dynamics, especially in allowing the capture of geodesics by isotropic singularities. As discussed in Sec. 4.1, the geodesic equations (written in terms of the affine parameter, not coordinate time) for the purely isotropic model can be mapped onto a Hamiltonian dynamics system. In this picture, it would naively seem that geodesic capture by a potential well would be impossible due to conservation of energy. Indeed, if one solves the geodesic equations in the affine parametrization, they will obtain solutions that pass right through the singularity in finite ‘proper time’ (measured in the affine parameter – proper time doesn’t technically exist for massless particles). When these geodesics are reparametrized in the global time coordinate, the point where the geodesic actually crosses the singularity is sent to infinite time. This is analogous to the well-known situation in GR where to an observer outside an event horizon, a light beam can never actually cross an event horizon – the actual collision is time-dilated to infinity. So conservation of energy in the affine parametrization is time-dilated away when we move to a description in terms of the global time coordinate.
In this paper we work in the other limit where we first reparametrize the geodesics equation in terms of t and only then solve for the geodesics. In this order of operations, we find that the friction terms in the reparametrized GE provide the ‘dissipation’ necessary for a geodesic to come to rest at a singularity.
We’ve clarified the relevant discussions in Secs. 3.3, 4.1, and 5.1.

6. Quoting S.A.J:

"How essential is the temporal-flatness condition in collimation property of nematic singularities and/or attractive property of isotropic singularities?"

The collimation property follows from the form of the geodesic equation in Eq.(33-35), in which temporal flatness has been used extensively to re-write the geodesic equation in terms of the disorder vectors (See Appendex A.)

While it seems possible to construct collimating singularities in a non-temporally flat spacetime, we don’t expect the geometric features found here to generalize.

7. Quoting S.A.J:

"The motivation for the geometrical models of velocity modulated Dirac equation comes from hight Tc superconducting compunds. Also there is a brief mention of graphene and twisted bilayer graphene. In these examples the gravitational (geometrical) coupling can be encoded into spatial-spatial compoents of the stress tensor."

It has been recently proposed in [arxiv:2108.08183] that in 8Pmmn borophene, substitution of boron atoms with carbon atoms substantially affects the tilt of the resulting Dirac cone. Since the tilting of the Dirac cone always induces a velocity anisotropy (like nemacity), how likely are these compounds to realize possible QGD in non-superconducting phase? This will correspond to modulation of the spatio-temporal components of the velocity. Does the random modulation of tilt velocity relax "temporal flatness" condition?

Indeed, tilted-cone scenarios, including 3D Weyl semimetals and the 8Pmmn borophene example mentioned provide examples of Hamiltonians in condensed matter that are equivalent to massless Dirac Fermions on curved-space manifolds. Further, these systems do satisfy the temporal flatness condition. If the tilt can be randomized by coupling to various forms of disorder or engineered through some set of control parameters, then these systems are very adjacent to our discussion.

However, the form of the geodesic equation we work with is particular to (2+1)-D metrics with spatial-spatial components and temporal flatness, so it isn’t clear if any of the geometry we find here generalizes to spacetime manifolds outside the scope we consider here. We leave a study of time-space mixing manifolds for future work.

We thank the referee for making us aware of this work. We have added a discussion of this class of materials to our introduction and included a citation to the paper in the comment.

8. Quoting S.A.J:

"All over sectoin 1.1 the concept of "Null geodesics" has been used repeatedly, which is a crucial concept to understand the paper. Perhaps it will help the readers to define it right at the beginning."

We have added a comment on the physical role of null geodesics to the introduction, at the first mention of the concept. We have also added a mathematical definition of geodesic mass to Sec. 3, immediately after the introduction of the geodesic equation.

9. Quoting S.A.J:
"Fig. 9B seems too crowded to follow examples of geodesics that collide or do not collide with the singularities. Maybe it helps the readers to make one curve of each category bolder than the others to assist the readers to follow at least two bold geodesics."

After some consideration of how to best clarify Fig.9B, we decided the best way was to add a clarifying comment in the caption. All geodesics are captured for launch angle in (-π/4, 3π/4), and all geodesics escape for launch angle in (3π/4, 7π/4). The confusion in the figure comes from the fact that for orbits launched near the critical angle, the decay/escape is very slow. Once the reader is explicitly told that all orbits are monotonically escaping/decaying, we think the figure is more clear.

References:
1. J. P. Dahlhaus, C.-Y. Hou, A. R. Akhmerov, and C. W. J. Beenakker, Geodesic scatter-
ing by surface deformations of a topological insulator, Phys. Rev. B 82, 085312 (2010),

---

## Round 2 · List of Changes

Please refer to "author comments" section, where changes are discussed alongside referee comments.

---

## Editorial Decision

published